# Towards Reliability of Parameter-free Optimization

## Abstract

Hyperparameter tuning, particularly the selection of an appropriate learning rate in adaptive gradient training methods, remains a challenge. To tackle this challenge, in this paper, we propose a novel parameter-free optimizer, ADAMG (Adam with the golden step size), designed to automatically adapt to diverse optimization problems without manual tuning. The core technique underlying ADAMG is our golden step size derived for the AdaGrad-Norm algorithm, which is expected to help AdaGrad-Norm preserve the tuning-free convergence and approximate the optimal step size in expectation w.r.t. various optimization scenarios. To better evaluate tuning-free performance, we propose a novel evaluation criterion, *reliability*, to comprehensively assess the efficacy of parameter-free optimizers in addition to classical performance criteria. Empirical results demonstrate that compared with other parameter-free baselines, ADAMG achieves superior performance, which is consistently on par with Adam using a manually tuned learning rate across various optimization tasks.

## 1 Introduction

Optimization serves as a foundation technique underpinning modern deep learning, which finds applications in various domains such as computer vision, AI for science, and natural language processing (Voulodimos et al., 2018; Redmon et al., 2016; Paul et al., 2021; Devlin et al., 2018; Radford et al., 2019). Some of the recent optimization approaches include embedding momentum mechanisms (Sutskever et al., 2013) and Adam (Kingma & Ba, 2014). Among them, adaptive gradient methods play an important role (Duchi et al., 2011; Kingma & Ba, 2014; Liu et al., 2023) due to their attractive performance across diverse problem structures, encompassing deep model architectures, data characteristics, and running hyper-parameters. Hyperparameter tuning associated with those optimization algorithms has a significant impact on the practical performance (Wilson et al., 2017). Especially, the Learning Rate (LR) tuning holds primary importance, since the precision of LR of popular (adaptive gradient) optimization methods is closely related to unknown problem properties, such as smoothness, error of estimated gradient, and the initial optimality gap. The close relationship between LR and these properties makes it a necessity to choose LR. Manual tuning is a commonly adopted practice for parameter selection, which requires computational resources and is prohibitive in large-scale machine-learning tasks.

Recently, there has been a growing interest in parameter-free training methods [1] due to their practical training efficiency and satisfactory performance. These methods are designed to eliminate the need for manual tuning parameters, achieving performance levels close to the best manually tuned training methods. Pioneering works in the realm of parameter-free training, incorporating mechanisms like subroutine&bisection subroutine mechanisms (Nesterov, 2015; Carmon & Hinder, 2022), are prohibitively expensive in the context of large-scale deep learning problems. This study directs its focus toward identifying parameter-free training methods that maintain comparable training costs to the most standard training algorithms, such as Adam (Kingma & Ba, 2014), for deep learning problems.

Current trends in parameter-free training methodologies are standard practices of embedding initial optimality gap into step size or drawing insights from the recently proposed DoG (Ivgi et al., 2023),

---

[1]In this manuscript, we will use the terms "parameter-free training methods" and "parameter-free optimizers" interchangeably.

which combines the classical results from AdaGrad-Norm step size and standard Gradient Descent (GD) step size (Duchi et al., 2011; Ward et al., 2020; Khaled et al., 2023). Existing approaches based on classical evaluation criteria demonstrate advantages in specific scenarios (Ivgi et al., 2023; Defazio & Mishchenko, 2023; Khaled et al., 2023; Mishchenko & Defazio, 2023). However, we observed that the state-of-the-art parameter-free optimizers exhibit unstable performance when confronted with diverse optimization problems, i.e., the performances of prior arts are sometimes much lower than the optimal manual tuning performance for some optimization tasks. This observation is obtained from the experimental results in Section 4.3, where a default choice like Adam(1e-3) with a cosine decay learning rate scheduler outperformed existing parameter-free optimizers in certain optimization scenarios. This prompts the following question: *How stable can a specific parameter-free optimizer be, achieving "close" performance to the manually-tuned best optimizer w.r.t. diverse optimization scenarios?*

To tackle this problem, we first explore how to systematically evaluate the effectiveness of parameter-free optimizers. Existing approaches mainly adopted the classical evaluation criteria, including *convergence speed* and *solution quality* (Kingma & Ba, 2014; Liu et al., 2023) for optimizers. However, in the context of parameter-free optimizers, limiting the validation to these two aspects has hindered researchers and engineers from confidently applying these optimizers to more complicated real-world tasks. Given a parameter-free optimizer is inherently expected to generalize to unseen optimization problems, it is critical to collectively measure how it consistently behaves across a spectrum of optimization problems. To this end, we introduce an additional novel evaluation criterion, *reliability*, for parameter-free training methods. This criterion evaluates whether a parameter-free optimizer consistently exhibits performances close to the best manually tuned optimizer across various optimization problems. Table 1 summarizes our experimental findings on reliability evaluation.

| DoG | DoWG | GOG | D-Adapt Adam | Prodigy Adam | *Adam(1e-3)* | (Proposed) ADAMG |
|------|------|------|------|------|------|------|
| Practical probability of achieving close performance of best manually tuning Adam. | | | | | | |
| 0.50 | 0.27 | 0.54 | 0.50 | 0.60 | 0.56 | **0.78** |
| Averaged Performance Gap with best manually tuning Adam. (%) | | | | | | |
| 8.0 | 12.2 | 8.2 | 11.2 | 5.8 | 11.4 | **3.0** |

Table 1: Practical probability of the parameter-free optimizers achieving close performance (less than 5% performance measure) with best manually tuning Adam and the averaged performance gap with best manually tuning Adam, which are derived from Table 4.

In this paper, we design a novel algorithm that leverages the ability of tuning-free convergence of AdaGrad-Norm (Duchi et al., 2011; Ward et al., 2020; McMahan & Streeter, 2010; Wang et al., 2023). Specifically, we formally define a golden step size of AdaGrad-Norm, drawing insights to preserve the ability of tuning-free convergence and approximate the optimal step size in expectation across various optimization problems. Subsequently, we derive the solution for the golden step size, which is independent of problem-specific properties, and integrate it into AdaGrad-Norm, resulting in our first parameter-free optimizer (Algorithm 1). By deeply integrating the derived golden step size with the Adam, we further introduce an Adam-like parameter-free method named ADAMG (Algorithm 2). Compared to existing parameter-free optimization methods, our proposed ADAMG stably outperforms all the baselines across various optimization tasks and achieves performance that closely aligns with the best performance achieved by manually tuning Adam.

We highlight the following contributions of the paper:

- We introduce a novel evaluation criterion, namely *reliability*, for assessing parameter-free training methods. Practical results show that this criterion reasonably validates the adaptability of parameter-free optimizers to diverse optimization problems.
- Based on our analysis of the classical AdaGrad-Norm algorithm, we propose the golden step size for AdaGrad-Norm, which is expected to preserve the ability of tuning-free convergence and approximate the optimal step size in expectation w.r.t. various optimization problems, resulting in an AdaGrad-Norm version parameter-free optimizer. Furthermore, we extend this concept to devise an Adam-like parameter-free method named ADAMG.

- Extensive experiments conducted on deep learning tasks reveal that ADAMG exhibits stable performance across a spectrum of optimization problems. Moreover, it closely aligns with the best performance achieved by manually tuning Adam, making it ready to be widely deployed.

## 2 RELATED WORK

Adaptive gradient methods have emerged as predominant choices for training deep learning optimization problems (Kingma & Ba, 2014; Balles & Hennig, 2018; Zhuang et al., 2020; Chen et al., 2023). Concurrently, there is a rising popularity of adaptive parameter-free approaches in the optimization landscape (Ivgi et al., 2023; Defazio & Mishchenko, 2023; Khaled et al., 2023; Mishchenko & Defazio, 2023).

The search mechanism is the natural avenue for achieving parameter-free capability. As we mentioned earlier, several works are the practice of search mechanism (Nesterov, 2015; Feurer & Hutter, 2019; Carmon & Hinder, 2022). Pioneering efforts without search mechanisms usually estimate the problem properties and are more concerned with guarantees for convex optimization problems. For instance, the Polyak step size schedule incorporates $f(\mathbf{x}_k) - f^\star$ into gradient descent for convex optimization problems (Polyak, 1987). The subsequent adaptations of this approach demonstrate fair performance in handling nonconvex problems (Loizou et al., 2021; Malitsky & Mishchenko, 2019; Latafat et al., 2023). Contrary to problem properties estimation, approaches adopted from online learning with theoretical guarantees, such as coin betting schemes, have been applied in deep learning optimization problems (Orabona & Pál, 2016; Orabona & Tommasi, 2017; Chen et al., 2022). A more recent trend involves the utilization of the initial optimality gap and the sum of gradient norm along training trajectory (over $K$ steps), $\frac{\max_{i \leq K} ||\mathbf{x}_i - \mathbf{x}_0|| \xrightarrow{\text{Approx.}} ||\mathbf{x}_0 - \mathbf{x}^\star||}{\sqrt{\sum_{i=1}^{K} ||\mathbf{g}_i||^2}}$ where $\mathbf{x}$ denotes parameters and $\mathbf{g}$ denotes stochastic gradient, as a means to adapt to unknown properties associated with gradient norms, smoothness, and strong convexity (Ivgi et al., 2023). The initial optimality gap $||\mathbf{x}_i - \mathbf{x}_0||$ primarily draws from classical results using gradient descent for the convex problems, while the gradient norm $\frac{1}{\sqrt{\sum_{i=1}^{K} ||\mathbf{g}_i||^2}}$ is inspired by AdaGrad (Bubeck et al., 2015; Nesterov et al., 2018; Duchi et al., 2011). However, the combination itself lacks convincing theoretical guarantees over nonconvex problems. Several works following this line of thought propose distance measure variants or integrate those techniques with Adam (Defazio & Mishchenko, 2023; Khaled et al., 2023; Mishchenko & Defazio, 2023).

## 3 METHOD

In Section 3.2, we start with analyzing and discussing the selection of LR that preserves the ability of tuning-free convergence of AdaGrad-Norm (Duchi et al., 2011; McMahan & Streeter, 2010; Wang et al., 2023; Faw et al., 2023). Then, incorporating the ability with the classical result of the descent lemma of smooth function, coupled with our idea about optimizing the solution across various optimization problems, we formulate and derive the corresponding solution for the *golden step size* of AdaGrad-Norm. This golden step size is expected to help AdaGrad-Norm *converge without tuning and approximate the optimal step size over various settings*. Finally, we discuss the scale-free property of the golden step size. These insights serve as the foundational principles for the development of our parameter-free optimizers, detailed in Section 3.5 and Section 3.6.

### 3.1 PRELIMINARY

We work on a differentiable (including non-convex) function $f(\cdot) : \mathbb{R}^d \to \mathbb{R}$ with the standard Euclidean norm $|| \cdot ||$. We follow the standard assumptions on function and stochastic gradient as Wang et al. (2023).

**Assumption 3.1** (*L*-smooth condition). *We assume that for any model parameter* $\mathbf{x}_1, \mathbf{x}_2$, $f$ *is differentiable and L-gradient Lipschitz such that* $||\nabla f(\mathbf{x}_1) - \nabla f(\mathbf{x}_2)|| \leq L||\mathbf{x}_1 - \mathbf{x}_2||$.

**Assumption 3.2** (Affine noise variance). *We assume that there exist positive constant $D_0$ and $D_1$ such that* $\mathbb{E}_{\mathcal{F}_k}[||\mathbf{g}_k||^2] \leq D_0 + D_1||\nabla f(\mathbf{x}_k)||^2, \ \forall k \geq 1$.

$\mathcal{F}_k = \sigma(g_{k-1}, \cdots, g_1)$ is the standard stochastic operator and stands for the sigma field of historical gradients up to $k - 1$.

## 3.2 GOLDEN STEP SIZE FOR ADAGRAD-NORM

AdaGrad-Norm converges when optimizing non-convex objectives under affine noise variance and bounded smoothness assumptions (Wang et al., 2023; Faw et al., 2023). Additionally, It enjoys the ability of tuning-free convergence, wherein differences in initial learning rates solely impact practical convergence speed rather than the final convergence. This attribute is considered a primary advantage inherited by subsequent variants. We initiate our analysis with the following corollary, which serves as the foundation of analyzing the preservation of the tuning-free convergence ability of AdaGrad-Norm (c.f. Algorithm 1 without the highlighted content).

**Corollary 3.3** (A simple variant of Thm. 2 in Wang et al. (2023)). *Given Assumptions 3.1 and 3.2, for AdaGrad-Norm with any learning rate $\eta > 0$, we have in expectation that:*

$$\min_{k \in [K]} ||\nabla f(\mathbf{x}_k)||^2 \leq \frac{1}{\mathcal{O}(\sqrt{v_K})} \left( \frac{4}{\eta} (f(\mathbf{x}_1) - f^\star) + 2D_1 \xi(0) \right.$$
$$\left. + \left( 2(L\eta D_1)^2 + D_1(L\eta)^2 + \frac{1}{2} D_0 \right) \frac{4}{\sqrt{v_0}} + 2L\eta \ln v_K \right),$$

*where $K$ denotes total steps, and $v_K$ is accumulated sum of the squared gradient norm (see Algorithm 1).*

The proof is presented in appendix C. Consider right-hand side of Cor. 3.3 as a function w.r.t. $\eta$: $h(\eta) := \frac{1}{\sqrt{v_K}} \left( \gamma_1 \eta^2 + \gamma_2 \eta \ln v_K + \frac{\gamma_3}{\eta} \right)$, where $\gamma_1, \gamma_1$, and $\gamma_3$ denote the corresponding problem-dependent values for simplification purpose. We note that the accumulated gradient norm $v_K$ increases; therefore, $h(\eta)$ must be a decreasing sequence to achieve tuning-free convergence. We discuss two possible cases of $\eta$ preserving the tuning-free convergence ability:

- Case 1: Supposing $\eta$ is a constant value.
- Case 2: Supposing $\eta$ is constant but *dynamic* w.r.t. $K$, and one possible solution is $\eta = (v_K)^q$, where $0 < q < \frac{1}{4}$ and $v_K > 1$, so $h(\eta) = \gamma_1 (v_K)^{2q - \frac{1}{2}} + \gamma_2 (v_K)^{q - \frac{1}{2}} \ln v_K + \gamma_3 (v_K)^{-q - \frac{1}{2}}$ is continually decreasing with the increasing of $v_K$.

Simultaneously, considering the general updating step, which can be easily derived from the descent lemma of smooth function: $f(\mathbf{x}_{k+1}) \leq f(\mathbf{x}_k) - \eta_k \nabla f(\mathbf{x}_k) \mathbf{g}_k + \eta_k^2 L ||\mathbf{g}_k||^2$. Since the right-hand side of the descent lemma forms a quadratic function w.r.t $\eta$, the (worst-case) optimal progressive step-size $\eta^{\text{opt}} = \frac{1}{2} \eta^{\text{div}}$. Here, $\eta^{\text{div}}$ represents the step size that leads to divergence ($\forall \eta > \eta^{\text{div}}$ is diverging step size).

Incorporating the concepts of preserving tuning-free convergence and achieving $0.5\times$ diverging step size under various settings, we formally formulate the *golden step size* of AdaGrad-Norm as

$$\eta^{\text{gold}} = \frac{1}{2} \arg \max_{\eta} \mathbb{E}_{x \in \mathbb{R}^+} h(x, \eta) \text{ s.t. } \lim_{x \to +\infty} \left( h(x, \eta) = \frac{1}{\sqrt{x}} (\gamma_1 \eta^2 + \gamma_2 \eta \ln x + \frac{\gamma_3}{\eta}) \right) = 0, \quad (1)$$

where $x := v_K$ for simplification purposes. Here, the expectation over $x \in \mathbb{R}^+$ denotes various settings, the constraint $\lim_{x \to +\infty} h(x, \eta) = 0$ ensures reservation of tuning-free convergence, and $\arg \max_{\eta} h(x, \eta)$ approximates the step size that drives the objective function to diverge, i.e., the potentially largest $h(x, \eta)$. Please also refer to the discussion regarding incorporating the optimal progressive learning rate and the learning rate that converges with the training trajectory in Section 5.

## 3.3 SOLUTION OF THE GOLDEN STEP SIZE

We now provide the analytical solution for equation 1. Firstly, we derive the domain of $\eta$ based on the constraint. Considering the constraint $\lim_{x \to +\infty} \frac{1}{\sqrt{x}} (\gamma_1 \eta^2 + \gamma_2 \eta \ln x + \frac{\gamma_3}{\eta}) = 0$, to ensure $\eta$ satisfies the constraint, one straightforward approach is considering $\lim_{x \to +\infty} \frac{\gamma_1 \eta^2}{\sqrt{x}} = 0$, $\lim_{x \to +\infty} \frac{\gamma_2 \eta \ln x}{\sqrt{x}} = 0$, and $\lim_{x \to +\infty} \frac{\gamma_3}{\eta \sqrt{x}} = 0$. This implies that the domain of $\eta$ is the in-

tersection of sub-domains where each sub-component $\{\frac{\gamma_1 \eta^2}{\sqrt{x}}, \frac{\gamma_2 \eta \ln x}{\sqrt{x}}, \frac{\gamma_3}{\eta \sqrt{x}}\}$ achieves 0 simultaneously. In other words, $\mathcal{O}(\eta) = \left(< \mathcal{O}(x^{\frac{1}{4}})\right) \cap \left(< \mathcal{O}(\frac{\sqrt{x}}{\ln x})\right) \cap \left(> \mathcal{O}(x^{-\frac{1}{2}})\right) = \left(< \mathcal{O}(x^{\frac{1}{4}})\right) \cap \left(> \mathcal{O}(x^{-\frac{1}{2}})\right) = \mathcal{O}(x^t)$ with $t \in (-\frac{1}{2}, \frac{1}{4})$. Therefore, we consider two cases as we discussed in Section 3.2: when $\eta := x^t$ where $t \in (-\frac{1}{2}, \frac{1}{4})$ (Case 2) and when $\eta$ is a constant value (Case 1), which covers the above domain. We then compare the maximum of the expectation $\max \mathbb{E}[h(x, \eta)]$ between these two cases.

Given $\eta := x^t$ where $t \in (-\frac{1}{2}, \frac{1}{4})$ (Case 2), and supposing $x$ is bounded and uniformly distributed, i.e., $x \sim \mathcal{U}(C_1, C_2)$, where $C_2 \gg C_1 > 1$, we have

$$\mathbb{E}_{x \sim \mathcal{U}(C_1, C_2)} \frac{1}{\sqrt{x}} \left(\gamma_1 \eta^2 + \gamma_2 \eta \ln x + \frac{\gamma_3}{\eta}\right) = \frac{1}{C_2 - C_1} \int_{C_1}^{C_2} \frac{1}{\sqrt{x}} (\gamma_1 x^{2t} + \gamma_2 x^t \ln x + \gamma_3 x^{-t}) dx$$

$$= \frac{1}{C_2 - C_1} \int_{C_{c1}}^{C_2} \gamma_1 x^{-\frac{1}{2}+2t} + \gamma_2 x^{-\frac{1}{2}+t} \ln x + \gamma_3 x^{-\frac{1}{2}-t} dx$$

$$= \frac{1}{C_2 - C_1} \left(\frac{\gamma_1}{\frac{1}{2}+2t} x^{\frac{1}{2}+2t} + \frac{\gamma_2}{\frac{1}{2}+t} x^{\frac{1}{2}+t} \ln x - \frac{\gamma_2}{(\frac{1}{2}+t)^2} x^{\frac{1}{2}+t} + \frac{\gamma_3}{\frac{1}{2}-t} x^{\frac{1}{2}-t}\right)\Big|_{C_1}^{C_2}$$

$$\approx \mathcal{O}\left(\frac{C_2^{\frac{1}{2}+2t} - C_1^{\frac{1}{2}+2t}}{C_2 - C_1}\right) \approx \mathcal{O}(C_2^{-\frac{1}{2}+2t}).$$

Since $t \in (-\frac{1}{2}, \frac{1}{4})$, it is straightforward to observe that $\eta = \lim_{t \to \frac{1}{4}^-} x^t$ attains highest expectation value with $\mathbb{E}_{x \sim (C_1, C_2)}[h(x, \eta)] \approx \mathcal{O}(C_2^{-\frac{1}{2}+2\frac{1}{4}}) = \mathcal{O}(1)$.

Given $\eta$ is constant value (Case 1), and supposing $x$ is bounded and uniformly distributed, i.e., $x \sim \mathcal{U}(C_1, C_2)$, where $C_2 \gg C_1 > 1$, we have

$$\mathbb{E}_{x \sim \mathcal{U}(C_1, C_2)} \frac{1}{\sqrt{x}} \left(\gamma_1 \eta^2 + \gamma_2 \eta \ln x + \frac{\gamma_3}{\eta}\right) = \frac{1}{C_2 - C_1} \int_{C_1}^{C_2} \frac{1}{\sqrt{x}} \left(\gamma_1 \eta^2 + \gamma_2 \eta \ln x + \frac{\gamma_3}{\eta}\right) dx$$

$$= \frac{1}{C_2 - C_1} \left(2(\gamma_1 \eta^2 + \frac{\gamma_3}{\eta})\sqrt{x}\Big|_{C_1}^{C_2} + \gamma_2 \eta(2\sqrt{x} \ln x - 4\sqrt{x})\Big|_{C_1}^{C_2}\right)$$

$$\approx \mathcal{O}\left(\frac{\ln(C_2) - \ln(C_1)}{\sqrt{C_2 - C_1}}\right) \approx \mathcal{O}\left(\frac{\ln(C_2)}{\sqrt{C_2}}\right).$$

With $C_2 \gg 1$, it follows that $\eta$ is a constant value attaining highest expectation value with $\mathbb{E}_{x \sim (C_1, C_2)}[h(x, \eta)] \approx \mathcal{O}(\frac{\ln(C_2)}{\sqrt{C_2}})$.

Since $\eta^{\text{gold}}$ desires the maximum expectation and $\mathcal{O}_{\text{Case 1}}(\frac{\ln(C_2)}{\sqrt{C_2}}) \ll \mathcal{O}_{\text{Case 2}}(1)$, we conclude that $\eta^{\text{gold}} = \frac{1}{2} \lim_{t \to \frac{1}{4}^-} x^t$, where $\to 1/4^-$ denotes approaching from the negative side, is the desired golden step size that achieves the maximum expectation under the defined constraint.

### 3.4 SCALE-FREE PROPERTY OF GOLDEN STEP SIZE

We adopted the definition of the scale-free property of an optimization method from Khaled et al. (2023), where it is defined as *multiplying $f$ by a constant factor $\alpha > 0$ and minimizing $\alpha f$ does not change the method's trajectory at all*. As mentioned in Section 1, the term parameter-free optimization refers to optimization algorithms devoid of tuning parameters, with scale-free being one of the preferred properties of parameter-free optimization methods (Khaled et al., 2023; Defazio & Mishchenko, 2023; Mishchenko & Defazio, 2023).

Taking Theorem 3.4 as an example, also appearing in Khaled et al. (2023); Yang et al. (2024), to illustrate the concept, Normalized Gradient Descent (NGD) is scale-free inherently, as rescaling $f$ to $\alpha f$ does not alter the step size trajectory, i.e., $\eta^{\text{opt}} = D_0 / \sqrt{K}$ remains unchanged before and after

rescaling. Meanwhile, if we can approximate $D_0$ dynamically, then NGD qualifies as parameter-free. We summarize the following key takeaways: 1). An immediate observation regarding scale-free methods is that the derived step size is not correlated with the scale $\alpha$. In terms of the parameter-free methods, the corresponding step size does not depend on terms such as the scale $\alpha$ or problem properties that are unknown or cannot be approximated; 2). It is important to note that parameter-free does not imply the ability to arbitrarily scale the derived step size.

Particularly, there is an immediate observation that $\eta^{\text{gold}}$ is independent of problem-dependent values $\gamma_1, \gamma_2$, and $\gamma_3$, further reinforcing the notion that rescaling function will not alter its trajectory.

**Theorem 3.4** (Example adopted from Levy (2017); Grimmer (2019)). *Suppose that $f$ is convex (bounded below by $f^\star := f(\mathbf{x}^\star)$) and satisfies Assumption 3.1. If we run NGD $\mathbf{x}_{k+1} = \mathbf{x}_k - \eta \frac{\nabla f(\mathbf{x})}{||\nabla f(\mathbf{x})||}$, we have $\min_{k=0,\cdots,K-1}(f(\mathbf{x}_k) - f^\star) \leq \frac{L}{2}(\frac{D_0^2}{2\eta K} + \frac{\eta}{2})^2$, where $D_0 := ||\mathbf{x}_0 - \mathbf{x}^\star||$, and $\eta^{opt} = \frac{D_0}{\sqrt{K}}$.*

### 3.5 ALGORITHM: ADAGRAD-NORM VERSION PARAMETER-FREE OPTIMIZER

Our analysis shows that when taking the *golden step size*, the updated AdaGrad-Norm algorithm is expected to preserve the ability of tuning-free convergence and approximate the optimal step size across various settings. We hereby propose a novel parameter-free optimization algorithm that integrates the golden step size into AdaGrad-Norm.

Since $\eta^{\text{gold}} = \frac{1}{2}\lim_{t \to \frac{1}{4}^-} x^t$, we define a numerator function $s(x) := px^q$, where $p \to 1/2, q \to 1/4^-$, to represent the embedding of the golden step size. The proposed parameter-free training method, named GOG (Golden step size over Gradients), is summarized in Algorithm 1. We highlight the modifications (highlighting in grey), involving the direct utilization of the numerator function as the additional coefficient of the previous parameter updating step. Note that Algorithm 1 approximates $v_K$ with $v_k$, and a similar idea is the optimality gap approximation exploited in the baseline method DoG and its variants (Ivgi et al., 2023). Due to page limitation, please refer to Appendix D.2 for discussions about the approximation.

---

**Algorithm 1** GOG based on AdaGrad-Norm

**Input:** initial parameter $\mathbf{x}_1$, step size $\eta_k$ (default 1), objective function $f(\mathbf{x})$, $p, q$

Initialize $v_1 = 0$

$s(x) = px^q$

**for** $k = 1$ **to** $K$ **do**

  $\mathbf{g}_k \in \partial f(\mathbf{x}_k, \xi_k)$

  $v_{k+1} = v_k + ||\mathbf{g}_k||^2$

  $r_{k+1} = s(v_{k+1})$

  $\mathbf{x}_{k+1} = \mathbf{x}_k - \eta_k \frac{r_{k+1}}{\sqrt{v_{k+1}}} \mathbf{g}_k$

**end for**

---

**Algorithm 2** ADAMG based on Adam

**Input:** initial parameter $x_1$, step size $\eta_k$ (default 1), $p, q$, $\beta_1, \beta_2$, $\beta_3$, $\epsilon$,

Initialize $m_1 = 0, v_1 = 0$, $r_1 = 0$

$s(x) = px^q$

**for** $k = 1$ **to** $K$ **do**

  $g_k \in \partial f(x_k, \xi_k)$

  $v_{k+1} = \beta_2 v_k + (1 - \beta_2)g_k^2$

  $\hat{v}_{k+1} = v_{k+1}/(1 - \beta_2^k)$

  $r_{k+1} = \beta_3 r_k + (1 - \beta_3)s(v_{k+1})$

  $m_{k+1} = \beta_1 m_k + (1 - \beta_1) r_{k+1} g_k$

  $\hat{m}_{k+1} = m_{k+1}/(1 - \beta_1^k)$

  $x_{k+1} = x_k - \min(\eta_k, 1/\sqrt{k}) \frac{\hat{m}_{k+1}}{(\sqrt{\hat{v}_{k+1}} + \epsilon)}$

**end for**

---

Note that $\beta_1, \beta_2, \beta_3, \epsilon$ in Algorithm 2 have default values of 0.95, 0.999, 0.95, and $10^{-8}$ respectively.

### 3.6 ALGORITHM: ADAM VERSION PARAMETER-FREE OPTIMIZER

Besides GOG, we further develop an Adam-like method incorporating the golden step size, leading to a practical parameter-free optimizer with momentum acceleration. Similarly, we use a numerator function $s(\cdot)$ for the embedding of the golden step size. We then approximate Adam's update of the Exponential Moving Average (EMA) w.r.t. the golden step size as follows: $r_{k+1} = \beta_3 r_k + (1 - \beta_3)s(v_{k+1})$, where $\beta_3 \in [0, 1)$ is the exponential decay rate.

Besides, inspired by D-Adapt Adam (Defazio & Mishchenko, 2023), we use EMA golden step size, $r_{k+1}$, in the gradient of first-moment estimation instead of the raw coefficient in the parameter updating step. Additionally, the term $1/\sqrt{k}$ in Algorithm 2 is a commonly adopted strategy appearing in optimizing stochastic problems against error caused by randomness in gradient estimation (Nesterov et al., 2018; Ge et al., 2015). We refer to it as a piratical practice. The parameter-free optimizer ADAMG, which embeds Golden step size with Adam, is summarized in Algorithm 2.

## 4 EXPERIMENTS

### 4.1 EVALUATION CRITERIA

The existing criteria for evaluating parameter-free approaches include convergence speed (E.g., loss curve) and solution quality (E.g., test accuracy), which are common in classical optimizer designs (Kingma & Ba, 2014; Liu et al., 2023) and parameter-free optimizer designs (Ivgi et al., 2023; Defazio & Mishchenko, 2023; Khaled et al., 2023; Mishchenko & Defazio, 2023). A good parameter-free optimizer consistently performs well across various optimization tasks, which is not covered by these criteria. We hereby introduce a novel criterion, *reliability*, to assess parameter-free optimizers.

Following, we formally introduce the definition of reliability to illustrate how to systematically evaluate the adaptability of a parameter-free training method to diverse optimization tasks, which is hard to achieve by observing the performance of single or independent tasks.

**Definition 4.1** (Reliability). *Given a set of optimization tasks, we initially group all the conducted optimization tasks into four categories based on the optimizers among Adam(1e-2), Adam(1e-3), Adam(1e-4), and Adam(1e-5), whichever yields the best performance measure, such as test accuracy. Reliability is calculated by averaging the practical ratio in each category, where the ratio is the statistical information about specific parameter-free optimizers achieving less than a 5% performance drop compared to the corresponding best Adam on all tasks in that category.*

### 4.2 SETUP

We compare ADAMG to DoG (Ivgi et al., 2023), DowG (Khaled et al., 2023), D-Adapt Adam (Defazio & Mishchenko, 2023), Prodigy Adam (Mishchenko & Defazio, 2023), and Adam(1e-3)&cosine LR scheduler with evaluation criteria reliability, solution quality and convergence speed. Unless otherwise specified, all Adam and Adam-type parameter-free optimizers are paired with a cosine learning rate scheduler. I.e., the default value of $\eta_k$ in ADAMG, D-Adapt Adam and Prodigy Adam is set to 1 with extra cosine annealing decay strategy, following the default choice of previous work (Defazio & Mishchenko, 2023; Mishchenko & Defazio, 2023). It is worth noting that our experiments in *Robustness against LR decay strategy* of Appendix A.4 show that the proposed ADAMG has little performance gap with or without LR decay. We adopt the same setting as previous work in our evaluation for a fair comparison.

The optimization tasks cover two main categories: Image tasks - full fine-tuning pretrained&randomly initialized DenseNet121 (Huang et al., 2017), ResNet18 (He et al., 2016), ViT-B/16 (Dosovitskiy et al., 2021), and VGG11 (Simonyan & Zisserman, 2014) under datasets CIFAR10, CIFAR100, and Tiny-ImageNet (Krizhevsky & Hinton, 2009; Russakovsky et al., 2015); Language tasks - full fine-tuning pre-trained BERT (Devlin et al., 2018) under GLUE benchmark and full fine-tuning LoRA on GPT2 (Radford et al., 2019; Hu et al., 2021) under GLUE benchmark (Wang et al., 2018). Note that we use the numerator function $s(x) = 0.2x^{0.24}$ for all optimization tasks, and the final formula slightly differs from our theoretical derivation, $p \to 1/2, q \to 1/4^-$, by a small coefficient (Please refer to Appendix D.1 for discussions about the scaling of the golden step size). Other setup details of the 38 tasks are summarized in Appendix A.1.

### 4.3 PERFORMANCE COMPARISON

The average performance measures for all 38 tasks are summarized in Table 2 and Table 3. A complete version for each table with standard deviation is provided in Appendix A.2 to satisfy the page length limitation. Please note that DenseNet, ResNet, ViT-B, and VGG refer to DenseNet121,

ResNet18, ViT-B/16, and VGG11, respectively. Below we review aggregated performance metrics derived from empirical studies in Table 2 and Table 3, including reliability, solution quality, and convergence speed.

**Reliability** naturally provides clearer insights into the effectiveness of the parameter-free optimizer across the pre-defined task categories. To evaluate reliability, we derive the statistical information from Table 2 and Table 3 about whether specific parameter-free optimizers achieve less than a 5% performance drop compared to the corresponding best Adam in each category. The results are presented in Table 4, showing that the proposed ADAMG exhibits the highest average reliability ratio.

It is worth noting that with the default 5% performance gap, the proposed method improves the reliability ratio from the second best of 0.60 to 0.78. Reliability ratio under 1% gap and 10% gap are provided in Table 10 of Appendix A.2, specifically, under a 1% gap, the reliability ratio improvement is from 0.52 to 0.70. Under a 10% gap, the reliability ratio improvement is from 0.67 to 0.83.

**Solution quality** is defined as $\frac{1}{n}\sum_{i=1}^{n} \max(\text{Perf}_i^{\text{best Adam}} - \text{Perf}_i^{\text{parameter-free}}, 0)$, where Perf represents performance measurements, such as test accuracy, for all $n$ optimization tasks. This metric indicates the average performance gap between the best Adam optimizer and the specific parameter-free optimizer. The results are presented in Table 4, highlighting that the proposed ADAMG achieves the best average solution quality.

**Convergence speed.** Figure 1, Figure 2, and Figure 3 in Appendix B.1 shows the loss curves of training pretrained&randomly initialized DenseNet121, ResNet18, ViT-B/16, and VGG11 under CIFAR10, CIFAR100, and Tiny-ImageNet. Figure 4 in Appendix B.2 shows the loss curves of full fine-tuning BERT and fine-tuning LoRA with GPT2 under selected tasks in the GLUE benchmark. In terms of convergence speed, the proposed ADAMG achieves competitive performance with the best optimizer across all the conducted optimization tasks.

**Performance of GOG.** Since acceleration techniques like momentum have been widely employed by modern optimizers and improve upon classical training methods such as SGD and AdaGrad-Norm by a large margin, we mainly discuss the results of the accelerated optimizers. Here we investigate optimizers without accelerations, comparing the proposed GOG and baselines DoG and DoWG. We see that the proposed GOG achieves the best performance in terms of reliability and solution quality, shown in Table 4.

## 5 CONCLUSION

In this work, we introduced a new mechanism for achieving parameter-free in adaptive gradient training methods by proposing the golden step size. This step size aims to preserve the tuning-free convergence and approximate the optimal step size in expectation w.r.t. various optimization problems. The resulting optimizer, ADAMG, demonstrates improved reliability and solution quality compared to previous methods, closely matching the performance of manually tuned Adam and facilitating deployment readiness.

**Limitation** 1). Despite the practical success, understanding the theoretical guarantees of the proposed approach is crucial. We discuss that while the proof framework for the convergence of AdaGrad-Norm in Wang et al. (2023) served as inspiration for our approach, it relies on the collective behavior of AdaGrad-Norm step size throughout the entire training trajectory. The diverging step size for AdaGrad-Norm is not directly connected to the one derived from the progressive updating formula, (Wang et al., 2023; Faw et al., 2023; Li et al., 2023), paving the way for future research. 2). While this work verifies that the proposed optimizer has a wide adaptability range through a broad spectrum of optimization tasks, it fails on some tail tasks possibly due to its expectation mechanism. Further investigations of embedding extra (approximated) problem properties such as the optimality gap may mitigate the issue but also lead to further work.

## ETHICS STATEMENT

Our work primarily focuses on theoretical and practical developments in optimization methods, which will enable efficient model training of deep model optimization tasks. However, we are also

| Dataset | Algorithm | Test accuracy (%)[a] | | | | | | | |
| | | Epoch 20&pre-trained network | | | | Epoch 100&randomly init. network | | | |
| | | DenseNet | ResNet | ViT-B | VGG | DenseNet | ResNet | ViT-B | VGG |
|---|---|---|---|---|---|---|---|---|---|
| CIFAR10 | Adam(1e-2) | 69.4 | 80.5 | 21.2 | 13.1 | **79.6** | **85.3** | 25.6 | 10.0 |
| | Adam(1e-3) | **88.1** | **92.4** | 75.8 | 84.5 | 75.4 | 84.9 | 36.4 | **77.3** |
| | Adam(1e-4) | 81.2 | 85.4 | **77.3** | **84.6** | 53.4 | 63.2 | **56.3** | 71.0 |
| | Adam(1e-5) | 64.3 | 72.3 | 57.9 | 77.2 | 48.2 | 58.3 | 29.3 | 61.8 |
| | DoG | $78.4^\times$ | 88.3 | $63.7^\times$ | 80.5 | $62.2^\times$ | $71.1^\times$ | 54.8 | 72.9 |
| | DoWG | $80.4^\times$ | $86.4^\times$ | $67.1^\times$ | 80.7 | $53.9^\times$ | $65.5^\times$ | $50.8^\times$ | $52.7^\times$ |
| | GOG | $79.5^\times$ | $85.7^\times$ | $68.6^\times$ | 82.6 | $54.7^\times$ | $66.3^\times$ | 53.7 | $66.3^\times$ |
| | D-Adapt Adam | 88.2 | 91.6 | 77.3 | $71.2^\times$ | $72.3^\times$ | 83.3 | $11.3^\times$ | $49.1^\times$ |
| | Prodigy Adam | 87.4 | 90.9 | 79.5 | 86.1 | $64.0^\times$ | $73.7^\times$ | $21.1^\times$ | 75.5 |
| | ADAMG | 86.1 | 91.1 | 78.6 | 87.3 | $68.1^\times$ | $75.9^\times$ | 58.1 | 77.4 |
| CIFAR100 | Adam(1e-2) | 37.3 | 45.0 | 7.3 | 1.0 | **47.2** | 52.1 | 8.4 | 1.0 |
| | Adam(1e-3) | **65.2** | **72.8** | 49.7 | 53.6 | 45.0 | **57.5** | 13.1 | 13.4 |
| | Adam(1e-4) | 55.7 | 62.3 | **51.1** | **60.1** | 23.2 | 36.4 | **27.8** | **33.5** |
| | Adam(1e-5) | 20.6 | 29.0 | 13.8 | 43.5 | 20.0 | 32.1 | 8.9 | 24.3 |
| | DoG | $50.6^\times$ | 69.0 | $30.7^\times$ | 56.4 | $33.6^\times$ | $47.4^\times$ | 29.2 | 31.8 |
| | DoWG | $55.7^\times$ | $65.3^\times$ | $40.4^\times$ | 56.2 | $26.7^\times$ | $38.1^\times$ | 24.9 | $1.0^\times$ |
| | GOG | $54.2^\times$ | $61.7^\times$ | $35.7^\times$ | 56.8 | $25.0^\times$ | $33.7^\times$ | 27.2 | $25.0^\times$ |
| | D-Adapt Adam | 65.4 | 71.8 | 53.6 | $43.0^\times$ | $43.7^\times$ | 55.7 | $1.0^\times$ | 29.2 |
| | Prodigy Adam | 64.4 | 72.1 | 55.9 | 62.4 | $42.0^\times$ | 53.7 | $5.7^\times$ | 41.2 |
| | ADAMG | 62.6 | 70.4 | 54.5 | 63.1 | $35.4^\times$ | $44.9^\times$ | 31.5 | 42.1 |
| Tiny-ImageNet | Adam(1e-2) | 38.5 | 43.4 | 3.9 | 0.5 | 37.2 | 45.5 | 1.6 | 0.5 |
| | Adam(1e-3) | **62.9** | **63.0** | **57.3** | 12.1 | **39.2** | **50.9** | 7.4 | 16.9 |
| | Adam(1e-4) | 59.5 | 60.0 | 56.5 | **59.6** | 16.8 | 35.7 | **16.4** | **35.2** |
| | Adam(1e-5) | 35.2 | 24.8 | 20.9 | 51.3 | 16.0 | 24.9 | 10.9 | 22.0 |
| | DoG | 61.4 | 69.1 | $49.5^\times$ | 57.4 | 34.5 | $45.5^\times$ | 14.2 | $24.9^\times$ |
| | DoWG | 60.7 | 61.1 | $45.3^\times$ | 57.2 | $24.4^\times$ | $28.5^\times$ | 15.5 | $7.8^\times$ |
| | GOG | 60.4 | 58.4 | $41.2^\times$ | 58.4 | $22.4^\times$ | $31.1^\times$ | 17.2 | $20.4^\times$ |
| | D-Adapt Adam | 60.2 | 60.3 | 64.5 | $23.1^\times$ | 36.0 | 47.3 | $1.2^\times$ | $27.4^\times$ |
| | Prodigy Adam | 62.0 | 63.6 | 63.2 | 58.8 | 40.5 | 53.6 | $8.1^\times$ | 33.8 |
| | ADAMG | 62.7 | 64.2 | 60.2 | 59.7 | $26.3^\times$ | $39.0^\times$ | 16.9 | 35.9 |

[a] $\times$ denotes that the performance measure of the specific parameter-free optimizer is at least 5% lower than the best Adam, which is highlighted in bold.

Table 2: Test accuracy with CIFAR-10, CIFAR100, and Tiny-ImageNet under 3 different seeds.

aware that the advancements may have broader implications, some of which could potentially have negative social impacts, such as misuse of the method in malicious application developments.

| Model arc. | Algorithm | SST-2 | MRPC | QQP | MNLI | QNLI | RTE | WNLI |
|---|---|---|---|---|---|---|---|---|
| | | Acc. | Acc. | Acc. | Matched Acc. | Acc. | Acc. | Acc. |
| BERT | Adam(1e-2) | 50.3 | 56.1 | 54.4 | 33.0 | 49.5 | 49.1 | **52.1** |
| | Adam(1e-3) | 50.3 | 68.4 | 63.2 | 32.1 | 49.8 | 50.9 | 52.1 |
| | Adam(1e-4) | 77.1 | 81.6 | 54.4 | 77.7 | 85.3 | 63.9 | 47.4 |
| | Adam(1e-5) | **92.5** | **83.2** | **90.7** | **84.1** | **91.3** | **65.8** | 38.0 |
| | DoG | 91.4 | 74.3$^\times$ | 89.1 | 83.1 | 90.6 | 51.9$^\times$ | 57.3 |
| | DoWG | 74.8$^\times$ | 72.3$^\times$ | 79.5$^\times$ | 59.5$^\times$ | 74.68$^\times$ | 51.1$^\times$ | 52.1 |
| | GOG | 91.5 | 85.6 | 88.9 | 82.5 | 90.8 | 66.2 | 52.1 |
| | D-Adapt Adam | 76.6$^\times$ | 68.4$^\times$ | 63.2$^\times$ | 66.1$^\times$ | 73.9$^\times$ | 61.3 | 52.1 |
| | Prodigy Adam | 91.5 | 73.5$^\times$ | 90.4 | 83.1 | 90.8 | 65.8 | 46.5$^\times$ |
| | ADAMG | 90.9 | 81.5 | 90.4 | 83.9 | 89.8 | 65.2 | 52.1 |
| GPT2 with LoRA | Adam(1e-2) | 50.3 | 61.6 | 32.7 | 32.3 | 50.0 | 52.5 | **48.4** |
| | Adam(1e-3) | 88.1 | **76.1** | 67.3 | 75.6 | 82.4 | 60.0 | 42.3 |
| | Adam(1e-4) | **90.8** | 71.3 | **81.8** | **78.8** | **84.9** | **61.6** | 44.1 |
| | Adam(1e-5) | 88.1 | 66.3 | 77.1 | 72.8 | 79.8 | 51.5 | 47.9 |
| | DoG | 64.2$^\times$ | 67.9$^\times$ | 43.0$^\times$ | 43.8$^\times$ | 51.7$^\times$ | 50.1$^\times$ | 48.8 |
| | DoWG | 90.4 | 69.8$^\times$ | 77.4 | 72.8$^\times$ | 81.9 | 50.8$^\times$ | 46.9 |
| | GOG | 90.0 | 45.4$^\times$ | 77.2 | 73.7$^\times$ | 81.4 | 53.1$^\times$ | 52.1 |
| | D-Adapt Adam | 55.2$^\times$ | 58.5$^\times$ | 27.6$^\times$ | 32.8$^\times$ | 50.2$^\times$ | 50.8$^\times$ | 50.7 |
| | Prodigy Adam | 85.7$^\times$ | 68.6$^\times$ | 27.5$^\times$ | 33.1$^\times$ | 52.2$^\times$ | 50.9$^\times$ | 51.6 |
| | ADAMG | 90.9 | 72.5 | 80.8 | 78.8 | 86.0 | 58.0 | 49.8 |

Table 3: Performance of fine-tuning pre-trained BERT with GLUE benchmark & Epoch 3 under 3 different seeds.

| Metrics | Algorithm | Adam(1e-2) | Adam(1e-3) | Adam(1e-4) | Adam(1e-5) | Avg.[c] |
|---|---|---|---|---|---|---|
| Reliability ratio[a,b] | DoG | 2/5 | 6/12 | 7/15 | 4/6 | 0.50 |
| | DoWG | 2/5 | 2/12 | 8/15 | 0/6 | 0.27 |
| | GOG | 2/5 | 2/12 | 9/15 | 6/6 | 0.54 |
| | D-Adapt Adam | 4/5 | 10/12 | 3/15 | 1/6 | 0.50 |
| | Prodigy Adam | 1/5 | 11/12 | 7/15 | 5/6 | 0.60 |
| | *Adam(1e-3)* | 4/5 | 12/12 | 7/15 | 0/6 | 0.56 |
| | ADAMG | 2/5 | 9/12 | 15/15 | 6/6 | **0.78** |
| Solution quality | DoG | 9.0 | 6.2 | 12.4 | 4.5 | 8.0 |
| | DoWG | 13.5 | 11.2 | 8.3 | 15.9 | 12.2 |
| | GOG | 13.2 | 13.6 | 5.3 | 0.8 | 8.2 |
| | D-Adapt Adam | 2.6 | 5.1 | 20.9 | 16.4 | 11.2 |
| | Prodigy Adam | 7.6 | 1.5 | 12.2 | 2.1 | 5.8 |
| | *Adam(1e-3)* | 2.6 | 0.0 | 10.9 | 32.1 | 11.4 |
| | ADAMG | 6.5 | 4.1 | 0.3 | 1.0 | **3.0** |

[a] The denominators of each entity in a row denote that the numbers of best optimizer for each task count for Adam(1e-2), Adam(1e-3), Adam(1e-4), and Adam(1e-5) are 5, 12, 15, 6 regarding the total 38 tasks.

[b] Each entity, e.g., 2/5 denotes that the parameter-free optimizer has less than 5% performance drop compared to the corresponding best hand-tuning Adam for 2 tasks in all 5 tasks.

[c] The average operation considers an even task distribution over Adam optimizers.

Table 4: *Reliability* demonstrates the statistical property of parameter-free optimizers. *Solution quality* shows an average performance gap to the solution from the best manual tuned Adam.

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

## A  EXPERIMENT

### A.1  SETUP

For the image tasks, we used the same batch size of 1024 and input image sizes 32×32, 32×32, and 64×64 for CIAFR10, CIAFR100, and Tiny-ImageNet datasets respectively. For the language tasks, we used the same batch size of 32 for BERT and GPT2 tasks. In particular, we used rank = 4 for the LoRA in GPT2 tasks. The number of training epochs is particularly mentioned in the corresponding performance table.

**Computer resources** All the experiments can be run on a single NVIDIA RTX A5000 Graphics Card (24G). Each image task in one setting can be completed within 5 hours, using less than 15GB of GPU memory. Similarly, each language task in one setting can be completed within 8 hours, using less than 10GB of GPU memory.

### A.2  COMPLETED VERSION RESULTS

The (completed version) average performance measure of all 42 tasks are summarized in Table 5, Table 6, Table 7, Table 8, and Table 9.

| Algorithm | Test accuracy (%) under CIFAR10 | | | | | | | |
| | Epoch 20 & pre-trained network | | | | Epoch 100 & randomly initialized network | | | |
| | DenseNet121 | ResNet18 | ViT-B/16 | VGG11 | DenseNet121 | ResNet18 | ViT-B/16 | VGG11 |
|---|---|---|---|---|---|---|---|---|
| SGD-M(1e-2) | 82.5±0.2 | 88.2±0.5 | 71.4±0.6 | 85.5±0.1 | 65.2±0.2 | 71.2±0.8 | 57.9±0.2 | 73.8±1.1 |
| SGD-M(1e-3) | 73.7±0.0 | 74.5±0.8 | 52.7±0.6 | 77.2±0.0 | 50.6±0.5 | 63.1±0.4 | 52.4±0.5 | 50.0±1.0 |
| SGD-M(1e-4) | 45.8±0.4 | 43.1±1.5 | 26.6±0.3 | 60.2±0.5 | 39.2±0.6 | 37.9±0.6 | 32.5±0.7 | 12.7±2.2 |
| SGD(1e-2) | 73.8±0.1 | 75.0±0.8 | 52.9±1.3 | 77.3±0.1 | 51.5±0.3 | 63.4±0.7 | 50.7±0.3 | 46.9±0.6 |
| SGD(1e-3) | 46.7±0.2 | 43.6±1.5 | 27.2±0.6 | 60.7±0.5 | 39.1±0.6 | 38.1±0.7 | 32.5±0.6 | 12.3±1.8 |
| SGD(1e-4) | 14.5±0.4 | 14.9±0.5 | 11.1±0.1 | 34.0±1.1 | 21.6±0.8 | 21.1±1.1 | 23.9±1.0 | 10.3±0.4 |
| Adam(1e-2) | 69.4±5.5 | 80.5±2.6 | 21.2±8.0 | 13.1±4.5 | 79.6±1.1 | 85.3±0.7 | 25.6±2.7 | 10.0±0.0 |
| Adam(1e-3) | 88.1±0.2 | 92.4±0.4 | 75.8±0.7 | 84.5±0.7 | 75.4±0.5 | 84.9±0.2 | 36.4±4.3 | 77.3±0.5 |
| Adam(1e-4) | 81.2±0.1 | 85.4±0.3 | 77.3±0.4 | 84.6±0.2 | 53.4±0.2 | 63.2±1.0 | 56.3±0.7 | 71.0±0.1 |
| Adam(1e-5) | 64.3±0.1 | 72.3±0.9 | 57.9±0.6 | 77.2±0.1 | 48.2±0.4 | 58.3±0.6 | 29.3±0.3 | 61.8±0.3 |
| DoG | 78.4±0.8 | 88.3±0.7 | 63.7±0.7 | 80.5±1.8 | 62.2±0.2 | 71.1±0.7 | 54.8±0.4 | 72.9±0.2 |
| DoWG | 80.4±0.4 | 86.4±0.7 | 67.1±1.2 | 80.7±1.2 | 53.9±0.5 | 65.5±0.4 | 50.8±0.9 | 52.7±3.1 |
| GOG | 79.5±0.1 | 85.7±0.4 | 68.6±0.5 | 82.6±1.0 | 54.7±0.4 | 66.3±0.7 | 53.7±0.7 | 66.3±0.8 |
| D-Adapt Adam | 88.2±0.1 | 91.6±0.4 | 77.3±1.1 | 71.2±10.2 | 72.3±0.3 | 83.3±0.3 | 11.3±1.2 | 49.1±27.7 |
| Prodigy Adam | 87.4±0.1 | 90.9±0.5 | 79.5±0.2 | 86.1±0.2 | 64.0±0.6 | 73.7±0.1 | 21.1±8.2 | 75.5±0.6 |
| ADAMG | 86.1±0.3 | 91.1±0.4 | 78.6±0.4 | 87.3±0.0 | 68.1±0.6 | 75.9±0.6 | 58.1±0.3 | 77.4±0.4 |

Table 5: Test accuracy with CIFAR-10 under 3 different seeds.

| Algorithm | Test accuracy (%) under CIFAR-100 | | | | | | | |
| | Epoch 20 & pre-trained network | | | | Epoch 100 & randomly initialized network | | | |
| | DenseNet121 | ResNet18 | ViT-B/16 | VGG11 | DenseNet121 | ResNet18 | ViT-B/16 | VGG11 |
|---|---|---|---|---|---|---|---|---|
| Adam(1e-2) | 37.3±6.3 | 45.0±1.1 | 7.3±1.1 | 1.0±0.0 | 47.2±1.5 | 52.1±0.7 | 8.4±3.7 | 1.0±0.0 |
| Adam(1e-3) | 65.2±0.2 | 72.8±0.8 | 49.7±2.5 | 53.6±1.6 | 45.0±0.4 | 57.5±0.4 | 13.1±1.7 | 13.4±17.5 |
| Adam(1e-4) | 55.7±0.1 | 62.3±0.7 | 51.1±0.4 | 60.1±0.3 | 23.2±0.3 | 36.4±1.0 | 27.8±0.3 | 33.5±0.5 |
| Adam(1e-5) | 20.6±0.4 | 29.0±0.5 | 13.8±0.5 | 43.5±0.2 | 20.0±0.2 | 32.1±0.5 | 8.9±0.3 | 24.3±0.2 |
| DoG | 50.6±2.5 | 69.0±2.7 | 30.7±2.7 | 56.4±0.2 | 33.6±0.3 | 47.4±0.7 | 29.2±0.3 | 31.8±1.2 |
| DoWG | 55.7±0.1 | 65.3±3.1 | 40.4±1.3 | 56.2±0.4 | 26.7±0.4 | 38.1±0.5 | 24.9±0.3 | 1.0±0.0 |
| GOG | 54.2±1.3 | 61.7±2.4 | 35.7±1.1 | 56.8±0.2 | 25.0±0.1 | 33.7±0.6 | 27.2±0.3 | 25.0±0.1 |
| D-Adapt Adam | 65.4±0.0 | 71.8±1.0 | 53.6±1.0 | 43.0±5.3 | 43.7±0.6 | 55.7±0.8 | 1.0±0.1 | 29.2±0.3 |
| Prodigy Adam | 64.4±0.1 | 72.1±0.8 | 55.9±0.3 | 62.4±0.5 | 42.0±0.2 | 53.7±0.7 | 5.7±1.6 | 41.2±0.6 |
| ADAMG | 62.6±0.2 | 70.4±1.3 | 54.5±0.1 | 63.1±0.1 | 35.4±0.0 | 44.9±0.4 | 31.5±0.2 | 42.1±0.4 |

Table 6: Test accuracy with CIFAR-100 under 3 different seeds.

### A.3  RELIABILITY RATIO UNDER DIFFERENT PERFORMANCE GAP

| Algorithm | Test accuracy (%) under Tiny-ImageNet | | | | | | | |
| | Epoch 20 & pre-trained network | | | | Epoch 100 & randomly initialized network | | | |
| | DenseNet121 | ResNet18 | ViT-B/16 | VGG11 | DenseNet121 | ResNet18 | ViT-B/16 | VGG11 |
|---|---|---|---|---|---|---|---|---|
| Adam(1e-2) | 38.5±0.6 | 43.4±0.5 | 3.9±2.0 | 0.5±0.0 | 37.2±0.9 | 45.5±0.2 | 1.6±0.9 | 0.5±0.0 |
| Adam(1e-3) | 62.9±0.2 | 63.0±0.2 | 57.3±0.4 | 12.1±16.5 | 39.2±0.1 | 50.9±0.5 | 7.4±0.9 | 16.9±11.6 |
| Adam(1e-4) | 59.5±0.3 | 60.0±1.4 | 56.5±0.2 | 59.6±0.2 | 16.8±0.2 | 35.7±0.4 | 16.4±0.2 | 35.2±1.0 |
| Adam(1e-5) | 35.2±0.2 | 24.8±0.8 | 20.9±0.1 | 51.3±0.2 | 16.0±0.3 | 24.9±0.3 | 10.9±0.2 | 22.0±0.4 |
| DoG | 61.4±0.3 | 69.1±2.0 | 49.5±0.9 | 57.4±1.7 | 34.5±0.3 | 45.5±0.5 | 14.2±0.3 | 24.9±1.0 |
| DoWG | 60.7±0.3 | 61.1±3.9 | 45.3±2.1 | 57.2±0.1 | 24.4±0.1 | 28.5±0.4 | 15.5±0.1 | 7.8±4.5 |
| GOG | 60.4±0.1 | 58.4±3.6 | 41.2±0.4 | 58.4±0.3 | 22.4±0.3 | 31.1±1.0 | 17.2±0.5 | 20.4±0.4 |
| D-Adapt Adam | 60.2±0.2 | 60.3±0.8 | 64.5±0.3 | 23.1±7.4 | 36.0±0.1 | 47.3±0.4 | 1.2±0.9 | 27.4±0.5 |
| Prodigy Adam | 62.0±0.2 | 63.6±1.0 | 63.2±0.3 | 58.8±0.0 | 40.5±0.4 | 53.6±0.4 | 8.1±1.1 | 33.8±0.1 |
| ADAMG | 62.7±0.1 | 64.2±1.4 | 60.2±0.3 | 59.7±0.3 | 26.3±0.4 | 39.0±0.2 | 16.9±0.1 | 35.9±0.4 |

Table 7: Test accuracy with Tiny-Imagenet under 3 different seeds.

| Algorithm | Fine-tuning pre-trained BERT under GLUE benchmark & Epoch 3 | | | | | | | | |
| | CoLA Matthews corr. | SST-2 Acc. | MRPC F1&Acc. | STS-B Pearson corr.& Spearman corr. | QQP F1&Acc. | MNLI Matched acc.& Mismatched acc. | QNLI Acc. | RTE Acc. | WNLI Acc. |
|---|---|---|---|---|---|---|---|---|---|
| Adam(1e-2) | 0.0±0.0 | 50.3±0.9 | 54.1±38.3&56.1±17.3 | nan | 17.9±25.4&54.4±12.4 | 33.0±1.7&33.0±1.6 | 49.5±0.0 | 49.1±2.6 | 52.1±6.0 |
| Adam(1e-3) | 0.0±0.0 | 50.3±0.9 | 81.2±0.0&68.4±0.0 | nan | 0.0±0.0&63.2±0.0 | 32.1±0.4&32.2±0.5 | 49.8±0.5 | 50.9±2.6 | 52.1±6.0 |
| Adam(1e-4) | 48.4±2.1 | 77.1±7.6 | 87.0±1.5&81.6±1.4 | 88.0±0.3&87.7±0.1 | 17.9±25.4&54.4±12.4 | 77.7±0.5&77.8±0.2 | 85.3±0.6 | 63.9±3.5 | 47.4±12.6 |
| Adam(1e-5) | 57.2±2.3 | 92.5±0.3 | 88.5±0.5&83.2±0.9 | 88.5±0.1&88.2±0.1 | 87.2±0.2&90.7±0.1 | 84.1±0.1&84.4±0.2 | 91.3±0.3 | 65.8±1.2 | 38.0±10.2 |
| DoG | 51.3±4.5 | 91.4±0.3 | 83.2±1.7&74.3±4.2 | 88.2±0.1&88.1±0.2 | 85.5±0.4&89.1±0.0 | 83.1±0.2&83.8±0.3 | 90.6±0.1 | 51.9±3.3 | 57.3±1.3 |
| DoWG | 17.5±24.7 | 74.8±17.3 | 82.3±1.8&72.3±2.6 | 88.1±0.1&88.3±0.1 | 55.8±39.4&79.5±11.5 | 59.5±20.6&60.3±21.1 | 74.6±17.8 | 51.1±2.7 | 52.1±6.0 |
| GOG | 53.8±3.7 | 91.5±0.3 | 89.7±0.4&85.6±0.1 | 88.5±0.3&88.5±0.2 | 85.1±0.2&88.9±0.1 | 82.5±0.2&83.3±0.2 | 90.8±0.2 | 66.2±2.4 | 52.1±6.0 |
| D-Adapt Adam | 0.0±0.0 | 76.6±18.1 | 81.2±0.0&68.4±0.0 | nan | 0.0±0.0&63.2±0.0 | 66.1±24.2&66.4±24.5 | 73.9±17.3 | 61.3±9.9 | 52.1±6.0 |
| Prodigy Adam | 53.5±3.3 | 91.5±1.3 | 82.0±5.4&73.5±7.5 | 57.6±39.2&57.0±39.6 | 87.3±0.1&90.4±0.2 | 83.1±0.5&83.6±0.6 | 90.8±0.1 | 65.8±3.5 | 46.5±13.9 |
| ADAMG | 50.6±3.2 | 90.9±0.4 | 87.0±2.1&81.5±3.3 | 88.7±0.6&88.5±0.6 | 87.1±0.1&90.4±0.0 | 83.9±0.4&84.3±0.1 | 89.8±0.3 | 65.2±3.5 | 52.1±6.0 |

Table 8: Performance of fine-tuning pre-trained BERT with GLUE benchmark under 3 different seeds.

| Algorithm | Fine-tuning LoRA on GPT2 under GLUE benchmark & Epoch 3 | | | | | | | | |
| | CoLA Matthews corr. | SST-2 Acc. | MRPC F1&Acc. | STS-B Pearson corr.& Spearman corr. | QQP F1&Acc. | MNLI Matched acc.& Mismatched acc. | QNLI Acc. | RTE Acc. | WNLI Acc. |
|---|---|---|---|---|---|---|---|---|---|
| Adam(1e-2) | 0.0±0.0 | 50.3±0.9 | 70.8±14.7&61.6±9.6 | 1.2±4.2&0.9±2.5 | 32.7±7.0&65.8±0.6 | 32.3±0.4&32.5±0.5 | 50.0±0.4 | 52.5±1.2 | 48.4±5.7 |
| Adam(1e-3) | 13.0±9.7 | 88.1±0.3 | 84.5±0.4&76.1±0.9 | 84.9±1.1&85.0±0.9 | 67.3±7.2&71.1±10.0 | 75.6±0.4&77.5±0.4 | 82.4±0.8 | 60.0±4.7 | 42.3±5.3 |
| Adam(1e-4) | 9.0±3.4 | 90.8±0.2 | 81.3±1.1&71.3±0.7 | 83.7±0.6&83.5±0.5 | 81.8±0.2&86.1±0.0 | 78.8±0.1&80.2±0.3 | 84.9±0.4 | 61.6±2.2 | 44.1±0.7 |
| Adam(1e-5) | 1.3±3.2 | 88.1±0.4 | 78.1±2.4&66.3±3.2 | 14.7±19.1&13.9±19.4 | 77.1±0.4&82.0±0.1 | 72.8±0.4&74.4±0.3 | 79.8±1.1 | 51.5±3.7 | 47.9±6.0 |
| DoG | 3.0±4.3 | 64.2±9.3 | 80.4±0.5&67.9±0.2 | -18.3±2.1&-17.6±2.4 | 43.0±19.8&67.8±2.0 | 43.8±0.0&45.4±0.4 | 51.7±2.0 | 50.1±1.3 | 48.8±5.4 |
| DoWG | 3.5±5.0 | 90.4±0.9 | 80.7±1.3&69.8±0.6 | 77.7±3.5&77.6±3.3 | 77.4±0.4&81.5±0.3 | 72.8±0.4&74.8±0.2 | 81.9±0.9 | 50.8±3.9 | 46.9±6.7 |
| GOG | 3.1±4.4 | 90.0±1.0 | 81.4±0.6&71.5±0.8 | 13.5±24.8&14.2±23.9 | 77.2±0.6&81.7±0.4 | 73.7±0.1&75.6±0.3 | 81.4±0.5 | 53.1±1.5 | 52.1±6.0 |
| D-Adapt Adam | 0.5±0.6 | 55.2±4.4 | 63.5±25.0&58.5±14.0 | 4.5±3.3&2.4±2.3 | 27.6±16.8&65.0±1.3 | 32.8±0.1&33.0±0.0 | 50.2±0.5 | 50.8±1.8 | 50.7±5.3 |
| Prodigy Adam | 0.0±0.0 | 85.7±1.9 | 81.0±0.3&68.6±0.3 | 35.1±35.3&33.8±36.0 | 27.5±21.1&64.6±1.0 | 33.1±0.5&33.2±0.3 | 52.2±1.4 | 50.9±2.8 | 51.6±5.7 |
| ADAMG | 24.2±5.0 | 90.9±0.6 | 82.6±0.6&72.5±1.4 | 83.9±0.5&83.6±0.6 | 80.8±0.4&85.6±0.1 | 78.8±0.1&79.9±0.2 | 86.0±0.5 | 58.0±4.9 | 49.8±5.8 |

Table 9: Performance of fine-tuning LoRA on GPT2 with GLUE benchmark under 3 different seeds.

| Metrics | Algorithm | Adam(1e-2) | Adam(1e-3) | Adam(1e-4) | Adam(1e-5) | Avg. |
|---|---|---|---|---|---|---|
| | DoG | 2/5 | 1/12 | 1/15 | 4/6 | 0.30 |
| | DoWG | 2/5 | 0/12 | 4/15 | 0/6 | 16.7 |
| | GOG | 2/5 | 0/12 | 3/15 | 4/6 | 0.31 |
| 1% Reliability ratio | D-Adapt Adam | 2/5 | 4/12 | 2/15 | 1/6 | 0.25 |
| | Prodigy Adam | /15 | 8/12 | 6/15 | 5/6 | 0.52 |
| | *Adam(1e-3)* | 2/5 | 12/12 | 5/15 | 0/6 | 0.43 |
| | ADAMG | 2/5 | 5/12 | 15/15 | 6/6 | 0.70 |
| | DoG | 2/5 | 9/12 | 13/15 | 6/6 | 0.57 |
| | DoWG | 2/5 | 6/12 | 9/15 | 0/6 | 0.37 |
| | GOG | 2/5 | 4/12 | 13/15 | 6/6 | 0.65 |
| 10% Reliability ratio | D-Adapt Adam | 5/5 | 10/12 | 4/15 | 1/6 | 0.56 |
| | Prodigy Adam | 2/5 | 11/12 | 8/15 | 5/6 | 0.67 |
| | *Adam(1e-3)* | 4/5 | 12/12 | 9/15 | 0/6 | 0.60 |
| | ADAMG | 3/5 | 9/12 | 15/15 | 6/6 | 0.83 |

Table 10: *Reliability* under 1% and 10% performance gap.

## A.4 Robustness against LR decay strategy

Recall that $\eta_k$ in ADAMG has a default value of 1 with an additional cosine annealing decay strategy, following the default choice of previous work Defazio & Mishchenko (2023); Mishchenko & Defazio (2023). Our empirical results show that ADAMG is robust to the decay strategy. Table 11 and Table 12 illustrate the performance difference between default ADAMG and modified ADAMG (with a constant $\eta_k$). The two methods show no noticeable performance gap.

| Algorithm | Test accuracy (%) under CIFAR10 | | | | | | | |
| | Epoch 20&pre-trained network | | | | Epoch 100&randomly initialized network | | | |
| | DenseNet | ResNet | ViT-B | VGG | DenseNet | ResNet | ViT-B | VGG |
|---|---|---|---|---|---|---|---|---|
| ADAMG | 86.1 | 91.1 | 78.6 | 87.3 | 68.1 | 75.9 | 58.1 | 77.4 |
| Modified ADAMG[a] | 85.6 | 91.1 | 77.6 | 86.7 | 68.3 | 75.6 | 57.1 | 76.1 |

[a] ADAMG with $\eta_k = 1$, which eliminates decay strategy.

Table 11: Test accuracy with CIFAR-10 under 3 different seeds.

| Algorithm | Fine-tuning pre-trained BERT under GLUE benchmark & Epoch 3 | | | | | | | | |
| | CoLA | SST-2 | MRPC | STS-B | QQP | MNLI | QNLI | RTE | WNLI |
| | Matthews corr. | Acc. | F1 | Pearson corr. | F1 | Matched acc. | Acc. | Acc. | Acc. |
|---|---|---|---|---|---|---|---|---|---|
| ADAMG | 50.6 | 90.9 | 87.0 | 88.7 | 87.1 | 83.9 | 89.8 | 65.2 | 52.1 |
| Modified ADAMG[a] | 49.9 | 90.3 | 87.7 | 88.5 | 87.1 | 83.5 | 90.0 | 67.3 | 52.1 |

[a] ADAMG with $\eta_k = 1$, which eliminates decay strategy.

Table 12: Performance of fine-tuning pre-trained BERT with GLUE benchmark under 3 different seeds.

# B TRAINING LOSS CURVES

## B.1 IMAGE TASKS

The training loss curves corresponding to Table 2 are demonstrated below.

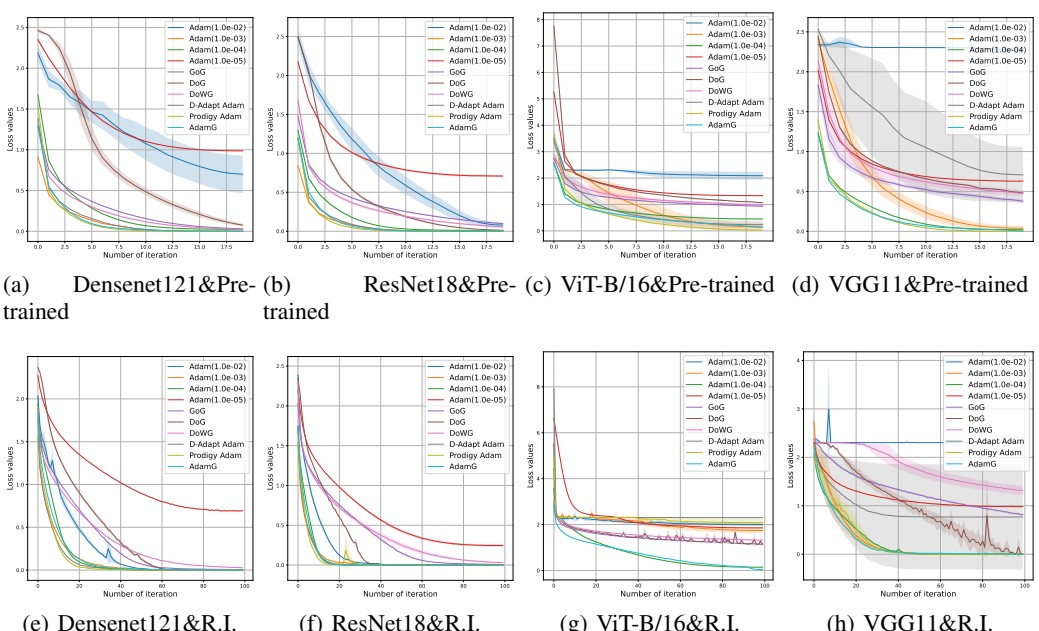

(a) Densenet121&Pre-trained (b) ResNet18&Pre-trained (c) ViT-B/16&Pre-trained (d) VGG11&Pre-trained

(e) Densenet121&R.I. (f) ResNet18&R.I. (g) ViT-B/16&R.I. (h) VGG11&R.I.

Figure 1: CIFAR10 experiments. Note Randomly Initialized (R.I.).

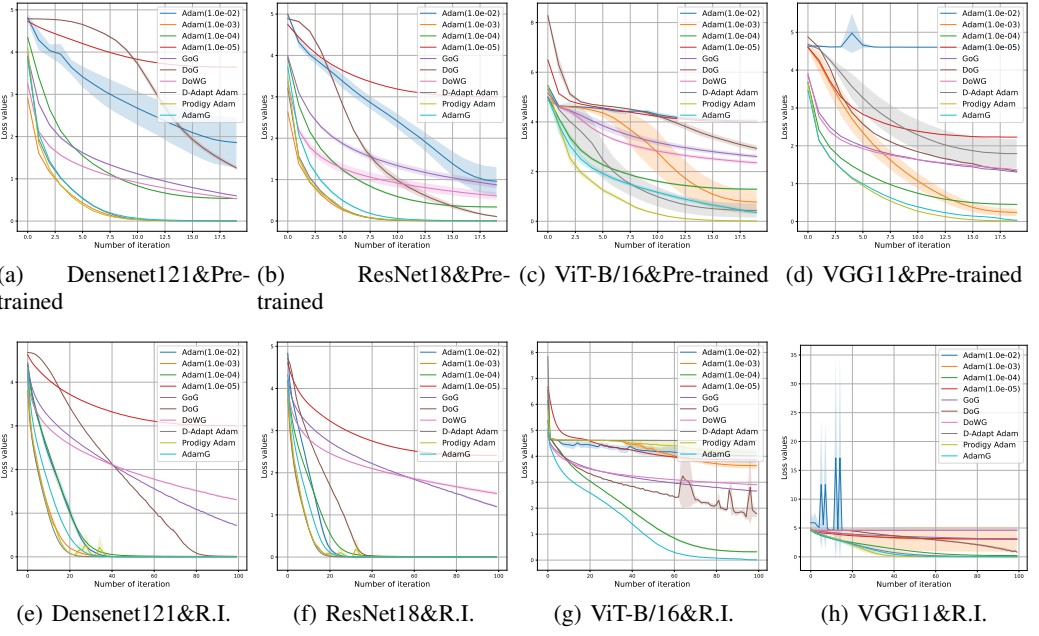

(a) Densenet121&Pre-trained (b) ResNet18&Pre-trained (c) ViT-B/16&Pre-trained (d) VGG11&Pre-trained

(e) Densenet121&R.I. (f) ResNet18&R.I. (g) ViT-B/16&R.I. (h) VGG11&R.I.

Figure 2: CIFAR100 experiments. Note Randomly Initialized (R.I.).

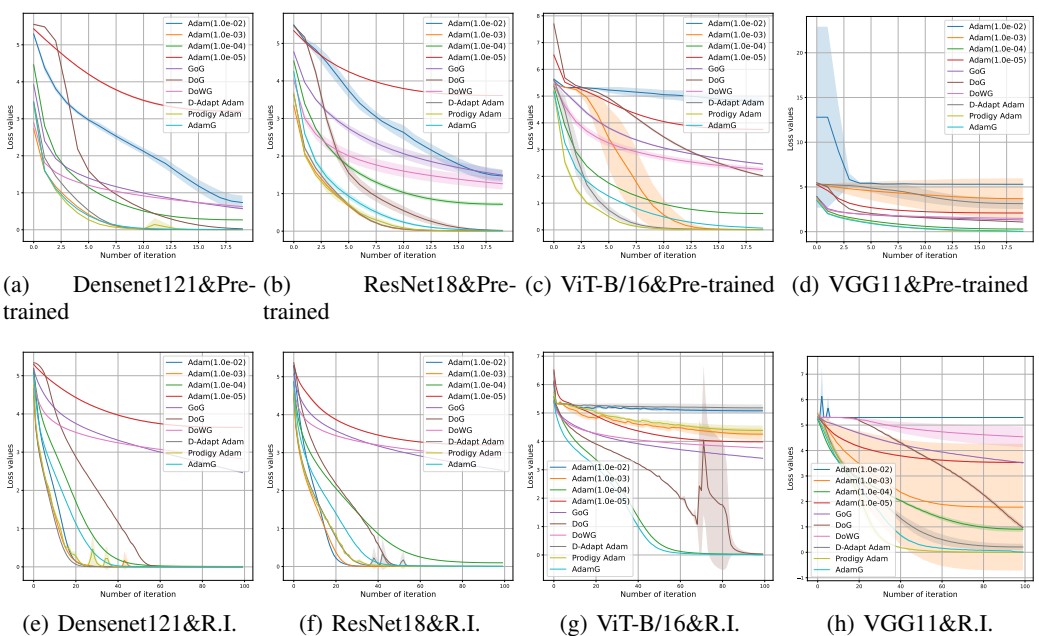

(a)      Densenet121&Pre-trained     (b)      ResNet18&Pre-trained     (c) ViT-B/16&Pre-trained     (d) VGG11&Pre-trained

(e) Densenet121&R.I.      (f) ResNet18&R.I.      (g) ViT-B/16&R.I.      (h) VGG11&R.I.

Figure 3: Tiny-ImageNet experiments. Note Randomly Initialized (R.I.).

## B.2 LANGUAGE TASKS

The training loss curves corresponding to above Table 3 are demonstrated below.

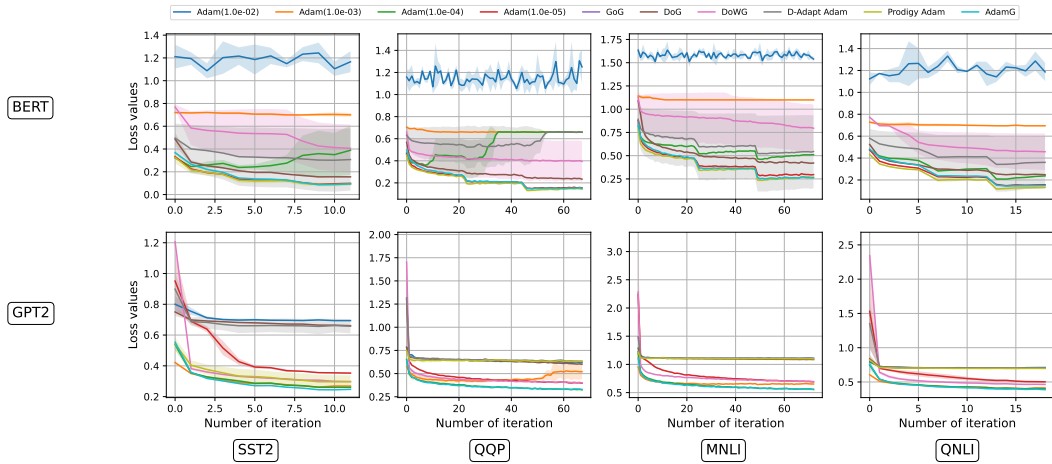

Figure 4: BERT and GPT2 under GLUE benchmark experiments.

## C    TECHNICAL PROOF DETAILS

**Corollary C.1** (a simple variant of Theorem 2 in Wang et al. (2023))**.** *Let Assumptions 1 and 2 hold. Then, for AdaGrad-Norm with any learning rate $\eta > 0$, we have in expectation that*

$$\min_{k \in [K]} ||\nabla f(\mathbf{x}_k)||^2 \leq \frac{1}{\mathcal{O}(\sqrt{v_K})} \Big( \frac{4}{\eta}(f(\mathbf{x}_1) - f^\star) + 2D_1\xi(0)$$

$$+ (2(L\eta D_1)^2 + D_1(L\eta)^2 + \frac{1}{2}D_0)\frac{4}{\sqrt{v_0}} + 2L\eta \ln v_K \Big).$$

*Proof.* We begin by following the inequality that is extracted from that of Theorem 2 of Wang et al. (2023).

$$\frac{1}{4}\eta \sum_{k=1}^{K} \mathbb{E}[\frac{||f(\mathbf{x}_k)||^2}{\sqrt{v_{k-1}}}] \leq f(\mathbf{x}_1) - \mathbb{E}[f(\mathbf{x}_K)] + \frac{\eta D_1}{2}\mathbb{E}[\xi(0) - \xi(T)] \tag{2}$$

$$+ (2\eta(L\eta D_1)^2 + \eta D_1(L\eta)^2 + \frac{\eta}{2}D_0)\frac{1}{\sqrt{v_0}} + \frac{L}{2}\eta^2(\mathbb{E}\ln v_K - \mathbb{E}\ln v_0). \tag{3}$$

Given the fact Wang et al. (2023):

$$\sum_{k=1}^{K} \mathbb{E}\left[\frac{||f(\mathbf{x}_k)||^2}{\sqrt{v_{k-1}}}\right] \geq \mathbb{E}\left[\frac{\sum_{k=1}^{K} ||\nabla f(\mathbf{x}_k)||^2}{\sqrt{v_K}}\right] \geq \frac{\mathbb{E}\left[\sqrt{\sum_{k=1}^{K} ||\nabla f(\mathbf{x}_k)||^2}\right]^2}{\mathbb{E}[\sqrt{v_K}]},$$

we have

$$\mathbb{E}\left[\sqrt{\sum_{k=1}^{K} ||\nabla f(\mathbf{x}_k)||^2}\right]^2 \leq \frac{4\mathbb{E}[\sqrt{v_K}]}{\eta}(f(\mathbf{x}_1) - \mathbb{E}[f(\mathbf{x}_K)]) + 2D_1\mathbb{E}[\sqrt{v_K}]\mathbb{E}[\xi(0) - \xi(T)]$$

$$+ (2(L\eta D_1)^2 + D_1(L\eta)^2 + \frac{1}{2}D_0)\frac{4\mathbb{E}[\sqrt{v_K}]}{\sqrt{v_0}}$$

$$+ 2L\mathbb{E}[\sqrt{v_K}]\eta(\mathbb{E}\ln v_K - \mathbb{E}\ln v_0).$$

Further applying the fact $\mathbb{E}[\frac{\sqrt{v_K}}{K}]$ is upper bounded by $\mathcal{O}(\frac{1}{\sqrt{K}})$ Wang et al. (2023), we have in expectation that:

$$\min_{k \in [K]} ||\nabla f(\mathbf{x}_k)||^2 \leq \frac{1}{\mathcal{O}(\sqrt{K})}\Big( \cdot \Big) \leq \frac{1}{\mathcal{O}(\sqrt{v_K})}\Big( \frac{4}{\eta}(f(\mathbf{x}_1) - f^\star) + 2D_1\xi(0)$$

$$+ (2(L\eta D_1)^2 + D_1(L\eta)^2 + \frac{1}{2}D_0)\frac{4}{\sqrt{v_0}} + 2L\eta \ln v_K \Big).$$

This concludes the proof. □

# D  DISCUSSION

## D.1  SCALING OF THE GOLDEN STEP SIZE

Our analysis in Section 3.3 derives $\eta^{\text{gold}} = \frac{1}{2}\lim_{t\to\frac{1}{4}^-} x^t$ and further suggests a numerator function $s(x) = px^q$, where $p \to 1/2, q \to 1/4^-$. Specifically, we employ $s(x) = 0.2x^{0.24}$ for all optimization tasks in the experiments. In the following, we verify the effectiveness of scaling $p$.

Table 13 and Table 14 demonstrate the performance comparisons between the default ADAMG and ADAMG (0.5), which employs $s(x) = 0.5x^{0.24}$. We observe that ADAMG (0.5) generally improves the performance on image tasks and fine-tuning LoRA on GPT2, but results in performance decay when fine-tuning BERT.

However, Table 15 provides clearer insights into the effectiveness of the scaling $p$ through reliability comparison. Compared with the default ADAMG, ADAMG (0.5) enhances adaptability on tasks that prefer a large LR Adam and reduces the adaptability on tasks that prefer a small LR Adam. Considering the inner expectation mechanism of the proposed method and the empirical performance, we believe that the scaling $p$ potentially shifts the covering but may not damage the range.

| Dataset | Algorithm | Test accuracy (%)[a] | | | | | | | |
| | | Epoch 20&pre-trained network | | | | Epoch 100&randomly init. network | | | |
| | | DenseNet | ResNet | ViT-B | VGG | DenseNet | ResNet | ViT-B | VGG |
|---|---|---|---|---|---|---|---|---|---|
| CIFAR10 | ADAMG | 86.1±0.3 | 91.1±0.4 | 78.6±0.4 | 87.3±0.0 | 68.1±0.6$^\times$ | 75.9±0.6$^\times$ | 58.1±0.3 | 77.4±0.4 |
| | ADAMG (0.5) | 87.4±0.2 | 92.9±0.6 | 79.6±0.2 | 87.4±0.5 | 74.5±0.4$^\times$ | 82.4±0.2 | 58.8±0.1 | 78.8±0.2 |
| CIFAR100 | ADAMG | 62.6±0.2 | 70.4±1.3 | 54.5±0.1 | 63.1±0.1 | 35.4±0.0$^\times$ | 44.9±0.4$^\times$ | 31.5±0.2 | 42.1±0.4 |
| | ADAMG (0.5) | 65.0±0.2 | 74.0±1.4 | 54.9±0.3 | 63.7±0.3 | 43.8±0.1 | 53.4±0.2 | 32.5±0.2 | 42.7±1.3 |
| Tiny-ImageNet | ADAMG | 62.7±0.1 | 64.2±1.4 | 60.2±0.3 | 59.7±0.3 | 26.3±0.4$^\times$ | 39.0±0.2$^\times$ | 16.9±0.1 | 35.9±0.4 |
| | ADAMG (0.5) | 63.6±0.2 | 65.4±1.3 | 64.1±0.5 | 57.7±0.0 | 34.8±0.1 | 45.4±0.3 | 17.9±0.3 | 33.7±0.6 |

Table 13: Test accuracy with CIFAR-10, CIFAR100, and Tiny-ImageNet under 3 different seeds.

| Algorithm | CoLA | SST-2 | MRPC | STS-B | QQP | MNLI | QNLI | RTE | WNLI |
| | Matthews corr. | Acc. Acc. | F1 F1 | Pearson corr. | F1 F1 | Matched acc. | Acc. Acc. | Acc. Acc. | Acc. Acc. |
|---|---|---|---|---|---|---|---|---|---|
| **BERT** | | | | | | | | | |
| ADAMG | 50.6±3.2$^\times$ | 90.9±0.4 | 87.0±2.1 | 88.7±0.6 | 87.1±0.1 | 83.9±0.4 | 89.8±0.3 | 65.2±3.5 | 52.1±6.0 |
| ADAMG (0.5) | 0.0±0.0$^\times$ | 50.9±0.0$^\times$ | 81.2±0.0$^\times$ | 26.3±29.5$^\times$ | 0.0±0.0$^\times$ | 36.6±6.1$^\times$ | 49.5±0.0$^\times$ | 50.9±2.6$^\times$ | 52.1±6.0 |
| **LoRA on GPT2** | | | | | | | | | |
| ADAMG | 24.2±5.0 | 90.9±0.6 | 82.6±0.6 | 83.9±0.5 | 80.8±0.4 | 78.8±0.1 | 86.0±0.5 | 58.0±4.9 | 49.8±5.8 |
| ADAMG (0.5) | 31.0±7.4 | 90.4±0.4 | 82.8±0.7 | 85.4±0.8 | 82.7±0.3 | 79.9±0.2 | 86.7±0.7 | 56.4±1.2 | 52.6±5.3 |

Table 14: Performance of fine-tuning pre-trained BERT with GLUE benchmark under 3 different seeds.

| Metrics | Algorithm | Adam(1e-2) | Adam(1e-3) | Adam(1e-4) | Adam(1e-5) | Avg. |
|---|---|---|---|---|---|---|
| Reliability ratio | ADAMG | 2/5 | 11/14 | 15/15 | 7/8 | **0.76** |
| | ADAMG (0.5) | 4/5 | 13/14 | 15/15 | 0/8 | 0.68 |

Table 15: *Reliability* ratio comparison, which is derived from Table 13 and Table 14, and provides clearer insights into the effectiveness of the scaling $p$. Compared with the default ADAMG, ADAMG (0.5) generally enhances adaptability on tasks that prefer a large LR Adam and reduces adaptability on tasks that prefer a small LR Adam. This potentially indicates a covering shift phenomenon.

## D.2 DYNAMIC STEP SIZE

In case 2, $\eta$ is constant but dynamic *with respect to K (upper-case)*, so $\eta := (v_K)^q$ can be considered as a constant value that obeys Corollary 3.3 and the analysis in Section 3.2. Finally, we approximates $v_K$ with $v_k$ in the proposed Algorithm 1. The approximation is intuitive and can be improved along with training. Besides, a similar idea is the optimality gap approximation, $\max_{i \leq K} ||x_i - x_0|| \xrightarrow{\text{Approx.}} ||\mathbf{x}_0 - \mathbf{x}^\star||$, exploited in the baseline method DoG and its variantsIvgi et al. (2023).

We further discuss dynamic $\eta$ *with respect to k (lower-case)*, i.e., $\eta := (v_k)^q$, which is naturally compatible with our algorithm design. We demonstrate that it aligns well our analysis framework with some wild assumptions and eliminates the need for approximation.

Particularly, we can update our Corollary 3.3 as a time-varying learning rate $\eta_k$, then, we have a similar form of conclusion, formulated as

$$\min_{k \in [K]} ||\nabla f(x_k)||^2 \leq \frac{1}{\mathcal{O}(\sqrt{v_K})} (c_1 \frac{\sum_{k=1}^{K} \eta_k^3}{\sum_{k=1}^{K} \eta_k} + c_2 \frac{\sum_{k=1}^{K} \eta_k^2}{\sum_{k=1}^{K} \eta_k} \ln v_K + c_3 \frac{1}{\sum_{k=1}^{K} \eta_k}),$$

where $c_1$, $c_2$, and $c_3$ are the corresponding coefficients. Ignoring the constant value and substituting $\eta_k$ with $(v_k)^q$, the right-hand side can be further reformulated as

$$\frac{1}{\sqrt{v_K}} \left( \frac{\sum_{k=1}^{K} (v_k)^{3q}}{\sum_{k=1}^{K} (v_k)^q} + \frac{\sum_{k=1}^{K} (v_k)^{2q}}{\sum_{k=1}^{K} (v_k)^q} \ln v_K + \frac{1}{\sum_{k=1}^{K} (v_k)^q} \right)$$

$$\rightarrow \mathcal{O} \left( \frac{1}{\sqrt{v_K}} (\frac{(v_K)^{3q+1}}{(v_K)^{q+1}} + \frac{(v_K)^{2q+1}}{(v_K)^{q+1}} \ln v_K + \frac{1}{(v_K)^{q+1}}) \right)$$

$$\rightarrow \mathcal{O} \left( \frac{1}{\sqrt{v_K}} ((v_K)^{2q} + (v_K)^q \ln v_K + \frac{1}{(v_K)^{q+1}}) \right).$$

Achieving a good approximation of the above first and second formulas necessitates that $\sum_{k=1}^{K} (v_k)^q \approx \frac{1}{q+1} (v_K)^{q+1}$, which imposes requirements on the sequence $\{v_1, \cdots, v_K\}$ (E.g., $\sum_{x=1}^{K} x^{\frac{1}{2}} \approx \int_1^K \frac{2}{3} x^{\frac{3}{2}}$) and leads to future investigations.

However, the desired result, as shown in the third formula above, will lead to the same level of expectation in Equation 1 as that derived from employing a constant step size (The difference in the last term does not affect the conclusion). This suggests that our algorithm design, $\eta_k = (v_k)^q$, is (possibly) naturally compatible with analysis in Section 3.2 and Section 3.3.

