# OpenReview forum: "Towards Reliability of Parameter-free Optimization"
_ICLR.cc/2025/Conference — Submitted to ICLR 2025_

### Official Review · Reviewer_Zvn8 · 2024-10-16

**Soundness:** 2
**Presentation:** 2
**Contribution:** 2
**Rating:** 5
**Confidence:** 3

**Summary:**

The paper proposes a new parameter-free version of Adam, using an automatic adaptation of the step size which they call golden step size. They also propose a "reliability" criterion to measure how good parameter-free optimizers are compared to the best performing Adam baseline.

**Strengths:**

The paper is clear and relatively well written. The idea of a reliability measure for optimizers is interesting and a decent contribution.

**Weaknesses:**

-    Very little analysis of the results or explanations of why AdamG seems to underperform on some of the more challenging training datasets like tiny imagenet, which leaves wondering how it would perform on even more challenging tasks such as full imagenet. Particularly surprising are its lack of performance with the ResNets.

-    While the method is parameter-free, they add additional parameters (like q=0.24) which are not properly analyzed or explained.

-    In many of the experiments, it would have been appropriate to run image classification for longer (200 or even 300 epochs is standard, especially when dealing with ViTs)

-    The paper leaves many of its results in the appendix, some of the main results are left at the bottom and not discussed or analyzed in depths.

-    Weight decay is completely left out and never mentioned in the paper. How does it compare when using Adam with weight decay? How does it compare with AdamW?

-    The figures in the appendix use full range on the Y axis, and make it close to impossible to see smaller differences. A zoom-in at the end or a smaller range on the Y axis would make it easier to compare them. Test accuracy plots are missing.

**Questions:**

-    How does AdamG compare when using Adam with weight decay, or with AdamW?
-    How does AdamG compare when running the image classification tasks until full convergence (~200/300 epochs for CIFAR-10/100)
-    How does the optimizer compare in wall-clock time to Adam?
-    How does AdamG perform on pretraining with small language modeling tasks, rather than finetuning only? 50-100K steps on GPT2 tiny would be enough.

---

> ### Author Response · Authors · 2024-11-21
>
> We thank the reviewer for the feedback. Below, we try to address the reviewer’s concerns.
>
> 1. $\textit{More challenging datasets and additional experiments}$ Our work includes a comprehensive evaluation across diverse image and language tasks and introduces a fair evaluation framework that can benefit the community. Regarding the computationally intensive experiment, such as the full ImageNet and large language model (LLM) pre-training, we plan to expand on this aspect upon acceptance.
>
> 2. $\textit{Additional parameters}$ Regarding the new hyperparameters, it is essential to clarify that the factors such as $q$ in the defined function are independent of problem properties, as established in our derivation. These factors do not resemble problem-dependent hyperparameters like LR.
>
> 3. $\textit{Weight decay and comparison with Adam}$ The main selected performance criterion is the performance improvement of the proposed method AdamG compared to other parameter-free optimizers. As demonstrated in the following table (Table 1 from the manuscript), the proposed method consistently outperforms baseline methods with significant margins:
>
> Table 1: Averaged Performance Gap with best manual-tuning Adam. (%)
> | DoG | DoWG | GOG | D-Adapt Adam | Prodigy Adam | AdamG |
> | --- | --- | --- | --- | --- |--- |
> | 8.0 | 12.2 | 8.2 | 11.2 | 5.8 | $\textbf{3.0}$ |
>
> Additionally, the ``best-tuned Adam'' including Adam with weight decay will not affect the conclusion in our main performance criterion. Below, we conducted additional experiments to verify the influence of Adam-type weight decay on the previously state-of-the-art and proposed methods. Specifically, the proposed method gains the most from tuning in every conducted task. Meanwhile, it also has relatively stabler performances.
>
> Table 2: Test accuracy with 100 epoch, randomly initialized network, CIFAR10 dataset under 3 different seeds. (%)
> | Algorithm | DenseNet121 | ResNet18 | ViT-B16 | VGG11 | Avg. Acc. |
> | --- | --- | --- | --- | --- | --- |
> | Prodigy | 64.0±0.6 | 73.7±0.1 | 21.1±8.2 | 75.5±0.6 | 58.5 |
> | Prodigy + 0.1 weight decay |10.0±0.0 | 19.6±0.2 | 10.0±0.0 | 10.0±0.0 | 12.4 |
> | Prodigy + 0.01 weight decay | 58.5±0.8 | 76.4±0.7 | 13.7±5.2 | 10.0±0.0 | 39.6 |
> | Prodigy + 0.001 weight decay | 77.1±0.8 | 85.7±0.7 | 51.6±0.7 | 10.0±0.0	56.1 |
> | AdamG | 68.1±0.6 | 75.9±0.6 | 58.1±0.3 | 77.4±0.4  | 69.8 |
> | AdamG + 0.1 weight decay  | 83.4±0.2  | 91.1±0.2 | 23.1±3.1 | 10.0±0.0 | 51.9 |
> | AdamG + 0.01 weight decay | 81.6±0.2 | 89.1±0.3 | 53.7±0.6 | 53.3±30.6 | 69.4 |
> | AdamG + 0.001 weight decay | 70.1±0.2 | 76.5±0.5 | 57.7±0.1 | 77.2±0.1 | 70.3 |

---

> > ### Comment · Reviewer_Zvn8 · 2024-11-21
> >
> > But how does standard AdamW compare to it? I know it's a lot to ask for rebuttal, but it would be interesting to see language modeling results. We find that often times some novel optimizer are quite overfitted to image classification, and underperform on language modeling, meaning that they are not very robust.
> >
> > Is there any indication how does AdamG compare in wall-clock time to standard Adam? Would be interesting in the table you sent to also have AdamW with the standard 1e-3 lr.

---

### Official Review · Reviewer_SQx1 · 2024-10-22

**Soundness:** 2
**Presentation:** 2
**Contribution:** 2
**Rating:** 3
**Confidence:** 3

**Summary:**

This paper introduces ADAMG, a new parameter-free optimizer that leverages a novel golden step size derived from the AdaGrad-Norm algorithm. Additionally, the authors propose a new evaluation criterion, "reliability", to assess how consistently parameter-free optimizers perform compared to manually tuned ones like Adam across various tasks. The empirical results suggest that ADAMG performs competitively with manually tuned Adam while retaining the benefits of being parameter-free.

**Strengths:**

1. **Novel Contribution**: The introduction of the golden step size for AdaGrad-Norm and its adaptation into ADAMG is an interesting and original theoretical advancement in parameter-free optimization.
2. **Comprehensive Evaluation**: The "reliability" criterion offers a new and effective way to assess parameter-free optimizers, going beyond traditional metrics like convergence speed and accuracy.
3. **Theoretical Foundation**: The paper is built on a solid mathematical framework, with a well-justified derivation of the golden step size and its properties.
4. **Practical Applications**: Empirical results show that ADAMG performs close to manually tuned Adam, which is beneficial for practitioners looking to avoid the time-consuming process of hyperparameter tuning.

**Weaknesses:**

See Questions.

**Questions:**

I am not an expert in parameter-free optimization, so my feedback might not be fully exhaustive. I won’t elaborate much on the theory—it looks sound at a glance. From my perspective, the aspect I want to highlight is that if you claim to outperform the best-tuned Adam, you need to be absolutely certain as this might have massive consequences.

Here are a few points:

1. Your experimental validation must be as strong as your claims: If you’re asserting that your method outperforms Adam (I suggest using AdamW, as it’s standard for LLMs), you must ensure you’ve thoroughly optimized its hyperparameters, schedulers, and other settings. This is because being better than a non-perfectly-optimized Adam(W) defeats the purpose of your effort. **I doubt this has been done properly**. To do this, I suggest taking two different LLM architectures (GPT and Llama) in three increasingly larger sizes (do not have to be hube but maybe 124, 248, 496M param) on three datasets of different quality (e.g. FineWeb,  Common Crawl...). Then, fine-tune AdamW to the best of your abilities: 5 learning rates, 5 batchsizes, 5 beta1, 5 beta2, 5 lambdas, and 3 different schedulers. I know this is very expensive and a huge burden, but once you do this, and you see that you performed consistently better (or even marginally less than AdamW, practitioners will pay attention to this paper).

2. The measure you propose is a beginning, but not maybe the best measure. A way to improve this is to devise measures that consider the "saved compute". For example, you can weight each experiment by how much compute they saved w.r.t. "best-tuned AdamW", or a normalized version of this. This way, there is no risk that some Param-Free optimizer receives a high score on tasks where fine-tuning Adam(W) is cheap, and thus "less valuable". Measuring success where it counts is important: The more compute you save, the higher the score.

3. After confirming that Adam(W) has been optimally tuned, you need to show that your optimizer consistently outperforms or at least matches Adam(W) in performance. You’ve already made some efforts in this direction and compared it against other parameter-free optimizers, which is good. **This has to be enhanced based on 1. and re-evaluated based on a measure in line with 2.**

4. I recommend shifting more of the empirical evaluation to LLMs, as they are a hot topic and where most fine-tuning resources are being spent.

5. Is your optimizer 4x more computationally expensive than SGD? If so, that makes it more expensive than AdamW, which is a significant drawback for LLMs. In this case, you might have to consider that your optimizer prevents the training of larger models, which will be a limitation.

I understand that addressing the points above might be too much to ask during rebuttal.

I will follow the discussion and reserve the right to modify my score based on further developments.

---

> ### Author Response · Authors · 2024-11-21
>
> We thank the reviewer for the feedback. Below, we try to address each of the reviewer’s concerns in detail.
> 1. $\textit{Experimental validation, Best-tuned Adam (or AdamW), and LLMs experiment}$ Thank you for raising this clarity issue. The performance evaluation in this work focuses on two primary criteria:\
> The main criterion is the performance improvement of the proposed method AdamG compared to other parameter-free optimizers. As demonstrated in the following table (Table 1 from the manuscript), the proposed method consistently outperforms baseline methods with significant margins:
>
> Table 1: Averaged Performance Gap with best manual-tuning Adam. (%)
> | DoG | DoWG | GOG | D-Adapt Adam | Prodigy Adam | AdamG |
> | --- | --- | --- | --- | --- |--- |
> | 8.0 | 12.2 | 8.2 | 11.2 | 5.8 | $\textbf{3.0}$ |
>
>
> On the other hand, the performance gap between the ``best-tuned Adam (or AdamW)’’ and the proposed method AdamG is also reported in this work, which is considered a complementary assessment. It is worth noting that a better approximated  `best-tuned Adam (or AdamW)’ does not affect the conclusion in our main performance criterion.
>
> Regarding the reviewer’s suggestion for additional experiments and measurements, particularly on computationally intensive tasks like large language model (LLM) training, we appreciate the valuable input. We plan to address this aspect in future work upon acceptance.
>
> 2. $\textit{Computational cost}$ Compared to Adam, our approach requires additional memory to store the momentum term $r$. However, it improves upon the previous state-of-the-art tuning-free method, Prodigy, by reducing the four-momentum terms down to three.

---

> > ### Comment · Reviewer_SQx1 · 2024-11-22
> > **Thanks**
> >
> > Dear Authors,
> >
> > Thank you for your reply.
> >
> > I will now refer to each of my points:
> >
> > **1.** The paper seems to highlight the fact that AdamG is on par with Adam: It is even in your abstract. However, you now highlight that what you are actually doing is that AdamG achieves better performance than other PF methods. While this seems to be true, the way you measure it and the set of experiments you do it on is somehow limited: You need to run these comparisons where it matters, on LLMs and large CV tasks. **While you promise to run these experiments upon acceptance, this is not how science works: First, you prove your claims with experiments, and then you get accepted. Not the other way around.**
> >
> > **2. and 3.**: These points were **completely disregarded** and not even commented on. Since I essentially **criticize** your "performance measure", and **propose** how to define another one that somehow measures some sort of cost saving or so, I would have expected at least a comment.
> >
> > **4**: See point 1.
> >
> > **5.**: I understand this point and it is a great strength of your method w.r.t. Prodigy or other PF methods. This does not address the issue that it is still more expensive than Adam and thus limits the actual deployment where it matters, i.e. **Large** language models. Maybe you could look into how "*Adam-mini: Use Fewer Learning Rates To Gain More*" reduces costs and see if you can build on that implementation trick.
> >
> > To conclude, I believe this is a promising direction, but it is still in its early stages and has room for improvement: I will keep my score.

---

### Official Review · Reviewer_yauh · 2024-11-02

**Soundness:** 2
**Presentation:** 3
**Contribution:** 2
**Rating:** 3
**Confidence:** 4

**Summary:**

This paper proposes a novel parameter-free optimization algorithm named AdamG. At the core of the proposed algorithm is the definition of “golden step size” derived aiming to approximate the optimal step size in expectation. Extensive experiment results are presented to demonstrate the effectiveness of the proposed optimization algorithm.

**Strengths:**

The idea of the proposed algorithm is very interesting.

The experimental results are fairly comprehensive.

**Weaknesses:**

I feel that there may be some important weaknesses in the current version of the paper.

First of all, unless I missed something very important, it seems that this paper does not provide any rigorous theory. The derivation in Section 3.3 relies on many 'approximately hold' equations, and Corollary 3.3 and Theorem 3.4 are not theoretical guarantees for the proposed AdamG, but known bounds for AdaGrad-Norm and NGD, respectively. In comparison, I find that the existing line of work on parameter-free optimization algorithms is highly theory-oriented. For example, the parameter-free algorithms in Table 1, such as DoG, DoWG, D-adaptation, and Prodigy, are all proposed with (and motivated by) rigorous theoretical guarantees. Therefore, not having rigorous theory for the proposed algorithm is a significant weakness and makes this paper diverge from the line of parameter-free optimization methods.

Second, the authors pointed out the first contribution of this paper is to propose the novel evaluation criterion named reliability. However, I am not convinced that reliability defined in Definition 4.1 is innovative. I believe it is a simple summary of what people do all the time – when proposing a new algorithm, it should be compared with existing algorithms with the optimally tuned hyperparameters. Moreover, the definition of reliability is not rigorous or general enough: why should the definition require specific comparison with Adam(1e-2), Adam(1e-3), Adam(1e-4), and Adam(1e-5)? Clearly, this definition hides assumptions on the (possible range of) scale of the optimal learning rate, is not mathematically reasonable, and is against the spirit of parameter-free optimization.

**Questions:**

Can you establish theroetical guarantees for the proposed algorithms?

How is the reliability evaluation criterion novel?

---

> ### Author Response · Authors · 2024-11-19
>
> We thank the reviewer for the feedback. Below, we try to address each of the reviewer’s concerns in detail.
>
> 1. $\textit{Theoretical guarantee}$ Similar to prior work on parameter-free optimization algorithms, our theoretical guarantees focus on a simpler algorithm, specifically AdaGrad-Norm in this study. This serves as the foundation for expanding to an Adam-type method, which does not currently include corresponding theoretical guarantees. A similar expansion can be observed in Section 5 of [1]. \
> We acknowledge that extending theoretical guarantees to the proposed AdamG is a valuable direction for future research. However, building on motivations derived from simpler settings or algorithms, such as convex problems in [1] or simpler methodologies in [2], is a standard and effective starting point for this line of work.
>
> 2. $\textit{Evaluation criterion}$ Thank you for raising this clarity issue. The customized categories of optimization tasks (e.g., Adam with learning rates 1e-2, 1e-3, 1e-4, and 1e-5) are motivated by two key considerations. First, in practical applications, the tuning process often involves classifying tasks into these categories. Second, different problem properties naturally suggest specific learning rates for convergence with a given optimization method. Thus, we believe an optimization method with diverse learning rates is representative of different optimization tasks. \
> Unlike previous work that conducts numerous isolated experiments, this study categorizes and consolidates all tasks into a single comprehensive table for more effective comparison.
>
> [1] Mishchenko, Konstantin, and Aaron Defazio. "Prodigy: An expeditiously adaptive parameter-free learner." arXiv preprint arXiv:2306.06101 (2023).
>
> [2] Maor Ivgi, Oliver Hinder, and Yair Carmon. DoG is SGD’s best friend: A parameter-free dynamic step size schedule. arXiv:2302.12022, 2023.

---

### Official Review · Reviewer_DtgK · 2024-11-08

**Soundness:** 2
**Presentation:** 2
**Contribution:** 2
**Rating:** 3
**Confidence:** 4

**Summary:**

The script proposed a new parameter-free method called AdamG. The design principle are derived from the analysis of Adagrad-norm.

**Strengths:**

The topic is important and interesting.

**Weaknesses:**

The motivation of the stepsize design is rather weak. The writing is sub-optimal. The efficacy of the proposed algorithm is not convincing. See below.

**Questions:**

**Weakness and questions:**

1. **The design principle is rather weak.** stepsize design is to optimize the constant terms in the convergence upper bound of AdaGrad-Norm. To me, this motivation is rather weak. Here are my reasons.

   1-1. the bound is derived based on worst case assumptions,which does not utilize much properties of neural nets.

   1-2. The upper bound itself is likely to be loose, similar as most existing convergence upper bound for adaptive methods. I don't think we can reach much insightful result by optimizing a loose upper bound of this kind.

   1-3. The derived upper bound is tailored for AdaGrad-Norm, which is significantly different from Adam. Such theory will miss many important properties of Adam. For instance, the update direction of AdaGrad-Norm is the same as SGD direction, but Adam is not.


2. **The algorithm description is not clear.**  In Algorithm 2, is the operation $r_{k+1}g_{k}$  operated coordinate-wisely? Also, how is eta_k changes with k? I did not find the update rule on eta_k.

4. **Extra memory overhead.** According to Algorithm2, do we need to store an additional momentum building block $r$? If yes, it would introduce significant  extra memory overhead. Please discuss.

5. **Extra hyper-parameter.** AdamG introduces a new hyperparameter beta3. Is AdamG sensitive to beta3? We need more ablation studies, otherwise, the proposed method is far from "parameter-free".

6. **The experiments are not convincing enough.** Currently, the results are only shown for small & middle-scaled experiments such as TinyImageNet and BLUE. It is much more convincing if the authors can provide evidence on LLM (>1B) pretraining.

---

> ### Author Response · Authors · 2024-11-19
>
> We thank the reviewer for the feedback. Below, we try to address each of the reviewer’s concerns in detail.
> 1. $\textit{Design principle}$ Similar to prior work on parameter-free optimization algorithms, our theoretical guarantees focus on a simpler algorithm, specifically AdaGrad-Norm in this study. This serves as the foundation for expanding to an Adam-type method, which does not currently include corresponding theoretical guarantees. A similar expansion can be observed in Section 5 of [1]. \
> We acknowledge that extending theoretical guarantees to the proposed AdamG is a valuable direction for future research. However, building on motivations derived from simpler settings or algorithms, such as convex problems in [1] or simpler methodologies in [2], is a standard and effective starting point for this line of work. \
> Regarding the worst-case assumption, it potentially suggests a broad applicability and forms a robust basis for our algorithm design.
>
> 2. $\textit{Algorithm description}$ Thank you for raising this clarity issue. In our approach $\eta_{k}$ represents the learning rate schedule, which we specify in section 4.2, Setup. Following standard practice, we set $\eta_{k}$ to 1 with an additional cosine annealing decay strategy, in line with previous work.
>
> 3. $\textit{Memory overhead}$ Compared to Adam, our approach requires additional memory to store the momentum term $r$. However, it improves upon the previous state-of-the-art tuning-free method, Prodigy, by reducing the four-momentum terms down to three.
>
> 4. $\textit{Hyper-parameters}$ Thank you for bringing this up. None of the tasks presented in Tables 2 and 3 require tuning, as all hyperparameters for AdamG are fixed.
>
> 5. $\textit{Experiments}$ Our work includes a comprehensive evaluation across diverse image and language tasks and introduces a fair evaluation framework that can benefit the community. Regarding the computationally intensive experiment on large language model (LLM) pre-training, we plan to expand on this aspect upon acceptance.
>
> [1] Mishchenko, Konstantin, and Aaron Defazio. "Prodigy: An expeditiously adaptive parameter-free learner." arXiv preprint arXiv:2306.06101 (2023).
>
> [2] Maor Ivgi, Oliver Hinder, and Yair Carmon. DoG is SGD’s best friend: A parameter-free dynamic step size schedule. arXiv:2302.12022, 2023.

---

> > ### Comment · Reviewer_DtgK · 2024-11-22
> > **Will keep my score**
> >
> > Thanks authors for the reply.
> >
> >  **As for design principle.** In my review comment, I raised the concern  on the motivation.  In particular, I think it seems rather weak to design methods based on a loose upper bound on Adagrad-norm (see my comment in 1-1, 1-2, 1-3.). The authors responded that what they did is "similar to prior work on parameter-free optimization algorithms" and discussed a lot on  "theoretical guarantee". I don't think this is actually related to my questions and my concern on the weak motivation remains unaddressed.
> >
> >
> > I will keep my score.

---

### Meta-Review · Area_Chair_ngUm · 2024-12-17

**Metareview:**

**Summary:** The paper introduces AdamG, a variant of Adam with a golden step size based on the AdaGrad-norm. It also introduces a reliability metric for evaluation.

**Strengths:** The reviewers agree that the problem that is considered is interesting and relevant to the community.

**Weaknesses:** In terms of design principles, the derivation of the step size is based on optimizing the constants in the bound, and the upper bound by itself might be loose. Additionally, derivations are based on AdaGrad norms, which are different from Adam's. The reviewers point out that the paper lacks a rigorous theoretical foundation, which is common for parameter-free optimization techniques. The reliability metric is also based on commonly used practices in optimization literature and therefore lacks novelty. Some reviewers point out the memory overhead of AdamG. The experiments are not sufficiently large and extensive.

**Decision:** At this stage, the paper has several shortcomings including the motivation for serving the step-size and experimental evaluation. I recommend a comparison to other methods using a standard benchmark such as [1]. For the current version, I recommend rejection.

[1] GE Dahl et al. "Benchmarking neural network training algorithms", arXiv preprint arXiv:2306.07179, 2023.

**Additional Comments On Reviewer Discussion:**

The authors address some of the concerns raised by the reviewers but do not provide additional experimental results. The paper requires further efforts to justify the design principles. The experimental results can be strengthened by showing improvements on a standard benchmark.

---

### Decision · Program_Chairs · 2025-01-22

Reject